# What Makes a Representation Good for Single-Cell Perturbation Prediction?

**Wenkang Jiang** [1]   **Yuhang Liu** [1 2]   **Yichao Cai** [1]   **Erdun Gao** [1 2]   **Jiayi Dong** [3]   **Ehsan Abbasnejad** [4]   **Lina Yao** [5 6]
**Javen Qinfeng Shi** [1 2]

## Abstract

Single-cell perturbation modeling is fundamental for understanding and predicting cellular responses to genetic perturbations. However, existing approaches, from causal representation learning to foundation models, often struggle with an overlooked challenge: gene expression is dominated by perturbation-invariant information, while perturbation-specific signals are intrinsically sparse. As a result, learned representations either entangle invariant and perturbation-specific information, leading to spurious and non-generalizable predictors, or suppress perturbation-specific signals altogether, rendering them ineffective for prediction. To address this, we propose *PerturbedVAE*, a general framework designed to resolve this signal imbalance. The framework explicitly separates perturbation-specific information from dominant invariant structure and recovers causal representations to effectively utilize such information for prediction. We further provide an identifiability analysis that characterizes the conditions under which sparse perturbation effects can be reliably recovered, thereby clarifying how the framework can be concretely specified under such conditions. Empirically, *PerturbedVAE* achieves state-of-the-art performance on a widely used benchmark across multiple evaluation settings, yielding significant gains on out-of-distribution combinatorial predictions and uncovering interpretable perturbation-response programs.

[1]Australian Institute for Machine Learning, Adelaide University, Australia [2]Responsible AI Research Centre, Australia [3]College of Computer Science and Artificial Intelligence, Fudan University, China [4]Department of Data Science and AI, Monash University, Australia [5]School of Computer Science and Engineering, University of New South Wales, Australia [6]CSIRO, Australia. Correspondence to: Yuhang Liu <yuhang.liu01@adelaide.edu.au>.

*Proceedings of the 43rd International Conference on Machine Learning*, Seoul, South Korea. PMLR 306, 2026. Copyright 2026 by the author(s).

## 1. Introduction

The goal of single-cell perturbation modeling is to predict how a cell's gene expression changes in response to genetic perturbations (Orgogozo et al., 2015), such as CRISPR-based perturbations (Jinek et al., 2012; Gilbert et al., 2014; Dixit et al., 2016; Replogle et al., 2020). Unlike standard predictive tasks, this problem involves perturbations that actively modify the underlying biological system. As a result, it presents two core challenges: generalization and interpretability. The first concerns generalization to unseen perturbations, e.g., combinatorial ones (Dixit et al., 2016; Replogle et al., 2020). The second concerns understanding how perturbations reshape underlying cellular programs (Huang & Liu, 2024; Lotfollahi et al., 2023).

Existing work on single-cell perturbation modeling has primarily focused on two directions: (i) learning causal representations, and (ii) learning general representations using foundation models (FMs) (See Sec. A for additional discussion). The first direction is motivated by leveraging causal mechanisms to support both generalization and interpretability (Lachapelle et al., 2022; Zhang et al., 2023; Lopez et al., 2022; de la Fuente et al., 2025). These approaches typically employ latent causal generative models to capture high-level causal variables that govern gene expression, and aim to recover such variables from data, also known as causal representation learning (Schölkopf et al., 2021; Liu et al., 2026c). The second direction is inspired by the success of FMs in various domains, and emphasizes data and model scale to learn powerful, transferable representations (Cui et al., 2024; Hao et al., 2024; Yang et al., 2022; Rosen et al., 2023; Theodoris et al., 2023).

**The Perturbation Suppression Hypothesis (Sec. 2.1).** Across both directions above, an empirical property of single-cell perturbation data is often underemphasized: perturbation-invariant information, such as background cellular programs, typically dominates gene expression, while perturbation-specific information is sparse. This gives rise to what we term the *Perturbation Suppression Hypothesis*: when invariant variation dominates the data, approaches that do not explicitly separate invariant and perturbation-specific information tend to fail systematically. In particular, such approaches either entangle in-

variant information into perturbation-related representations, undermining their perturbation-specific semantics and leading to poor generalization, as in existing causal representation learning methods, or suppress perturbation-induced signals that are critical for accurate perturbation prediction, as in FM-based approaches.

**Perturbation-Aware Representations (Sec. 2.2).** Motivated by the perturbation suppression hypothesis, we advocate for learning *perturbation-aware representations*, which explicitly aims to: 1) separate perturbation-specific information from dominant, perturbation-invariant one (*i.e., Extraction*), 2) organize the extracted perturbation-specific information in a manner that supports generalization, and crucially, interpretability (*i.e., Utilization*):

**Contributions.** Beyond introducing the perturbation suppression hypothesis and perturbation-aware representations (Sec. 2), we make the following contributions. First, guided by learning perturbation-aware representations, we propose a general framework, termed Perturbation-aware Variational Autoencoders (PerturbedVAE), which combines an alignment-based component to isolate perturbation-specific information from dominant, unperturbed cellular programs, with a latent causal model that organizes the isolated information, enabling principled generalization to unseen perturbations (Sec. 3). Second, we provide an identifiability analysis that characterizes the conditions under which perturbation-specific information can be reliably recovered. Importantly, this analysis clarifies how the proposed framework admits theoretically grounded specifications (Sec. 4). Third, we empirically demonstrate substantial improvements over strong baselines across multiple single-cell perturbation benchmarks, with particularly strong gains in generalization to unseen combinatorial perturbations, while yielding interpretable representations of perturbation effects (Sec. 5).

## 2. Perturbation Suppression and Beyond

We begin by introducing the perturbation suppression hypothesis, motivated by a fundamental yet often overlooked property of single-cell perturbation data: perturbation-invariant information typically dominates gene expression. From this perspective, we show that representations that do not explicitly distinguish perturbation-invariant from perturbation-specific information may fail to effectively utilize or preserve perturbation-induced signals. Finally, we introduce perturbation-aware representation learning to address this hypothesis.

### 2.1. Perturbation Suppression Hypothesis

Largely because of experimental, environmental, and cost constraints, perturbation-specific signals typically occupy only a small portion of the gene expression space, while unperturbed background programs account for the majority, even in large-scale perturbation datasets. For example, in widely used Perturb-seq benchmark datasets, the number of distinct perturbations is relatively limited (e.g., on the order of $\sim$284 in the dataset of Norman et al. (2019)), whereas gene expression profiles span tens of thousands of measured genes (e.g., $\sim$19,264 genes (Ahlmann-Eltze et al., 2025)). In this setting, methods that are not explicitly designed to separate perturbation-invariant and perturbation-specific information tend to exhibit the following failure modes: either the learned perturbation-related representations become entangled with invariant information, or perturbation-induced signals are suppressed, as below.

**Implications for Learning Causal Representations.** Although conceptually appealing, the effectiveness of learning causal representations is not closely supported by theoretical guarantees that are typically grounded in identifiability results, which ensure that the underlying causal variables can be recovered up to simple transformations. A key assumption underlying most existing identifiability results is access to sufficiently rich interventional data, i.e., that all latent causal variables are perturbed across environments (Lachapelle et al., 2022; Zhang et al., 2023; Lopez et al., 2022; de la Fuente et al., 2025).

In practice, however, single-cell perturbation data is often characterized by partial intervention: only a small subset of genes is perturbed, while large portions of genes remain invariant, as mentioned above. This mismatch between theoretical assumptions and available perturbation data in practice encourages invariant background information to be absorbed into perturbation-related (i.e., causal) representations. This may constitute perturbation suppression, whereby invariant information may be mixed into perturbation-related representations when such separation is not explicitly enforced. As a result, constructing causal models on such mixed representations may be inaccurate, leading to limited generalization ability.

**Implications for FMs.** FMs for single-cell transcriptomics are typically trained to learn generic representations by emphasizing fitting the overall data distribution. Given the dominance of perturbation-invariant information, such objectives naturally prioritize encoding shared, perturbation-invariant structure. As a consequence, perturbation-specific signals, which occupy only a small fraction of the expression space, may become less accessible in the learned representations.

To empirically assess this effect, we conduct a linear probing experiment on frozen representations learned by several FMs. Specifically, we treat perturbation conditions as surrogate labels (i.e., variable $\mathbf{u}$ defined in Sec. 3.1) and evaluate whether they are linearly decodable from the learned

representations, a standard diagnostic for assessing what information is encoded in an accessible form. We compare representations from UCE (Rosen et al., 2023), scFoundation (Hao et al., 2024), and Geneformer (Theodoris et al., 2023) against a Principal Component Analysis (PCA) baseline applied directly to gene expression.

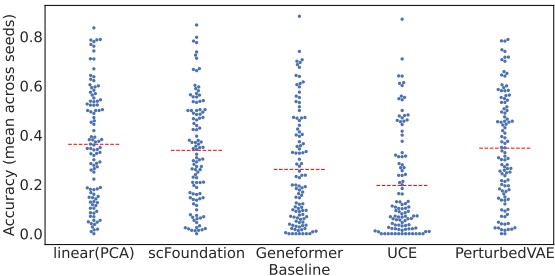

*Figure 1.* Linear probe accuracy (↑) for predicting perturbation labels. FM representations exhibit weaker linear decodability of perturbation labels than a PCA baseline, suggesting limited accessibility of perturbation-related information. In contrast, PerturbedVAE achieves competitive linear probing accuracy, indicating improved accessibility of perturbation-specific signals.

As shown in Fig. 1, representations learned by FMs exhibit weaker linear decodability of perturbation labels than a simple PCA baseline. Since PCA operates directly on the observed gene expression profiles, this result suggests that perturbation-related information is present in the data but becomes less readily accessible after being encoded by generic FM objectives. This observation provides empirical evidence consistent with the perturbation suppression effect in FM representations. In contrast, representations learned by the proposed PerturbedVAE yield competitive linear probing accuracy. See Sec. 5.2 for further results.

### 2.2. Learning Perturbation-Aware Representations

Taken together, the observations above suggest that the central challenge in single-cell perturbation prediction lies in how perturbation-specific information is extracted and utilized. Effective perturbation prediction requires representations that preserve sparse perturbation-induced signals, rather than suppressing them as in FM-based methods, while also preventing invariant structure from becoming entangled with perturbation-related representations, as can occur in learning causal representation methods. Moreover, such representations should organize perturbation-specific information in a form that supports generalization, typically for unseen and combinatorial perturbations.

This motivates us to learn perturbation-aware representations that *explicitly* extract perturbation-specific information from dominant invariant information, i.e., *extraction*, and organize the extracted information in a causally structured representation space for prediction under unseen perturbations, i.e., *utilization*.

## 3. PerturbedVAE: A Heuristic Framework for General Perturbation-Aware Modeling

Motivated by the notion of *perturbation-aware representations* introduced above, we present PerturbedVAE as a conceptual modeling framework that instantiates the design principles of *Explicit Extraction* and *Effective Utilization*. We emphasize that this section introduces the framework at a high level, motivated by the preceding analysis, rather than a fully specified implementation.

We will study this framework in Sec. 4 from a theoretical perspective and derive conditions under which perturbation-specific information can be reliably identified and utilized. These theoretical insights then guide the concrete instantiation of the framework, ensuring that the resulting model is grounded in principled guarantees.

### 3.1. A Latent Causal Generative Model

We begin by introducing the underlying causal generative model, which explicitly formulates perturbation-invariant background programs and perturbation-specific factors, and how they jointly give rise to the observed gene expression data. This model serves as the foundation for designing the proposed framework.

Figure 2 illustrates the proposed latent causal generative model for single-cell perturbation data. We assume that the observed gene expression $\mathbf{x}$ is generated from latent variables $\mathbf{z}$ through an unknown nonlinear mapping. To formulate genetic perturbations, we introduce a surrogate variable $\mathbf{u}$ that indexes the applied perturbation label (e.g., a one-hot encoding). Importantly, we do not assume access to the underlying biochemical intervention mechanism; it suffices to observe which perturbation condition is applied.

To reflect the distinction between unperturbed background cellular programs and perturbation-induced effects, we decompose the latent space into two components:

- $\mathbf{z}_\iota$ (*perturbation-invariant variables*), supported on $\mathcal{Z}_\iota \subseteq \mathbb{R}^{d_\iota}$, represents latent factors that remain stable across perturbation conditions and capture invariant background programs shared by all cells.

- $\mathbf{z}_\nu$ (*perturbation-responsive variables*), supported on $\mathcal{Z}_\nu \subseteq \mathbb{R}^{d_\nu}$, represents latent factors whose values change in response to genetic perturbations and encode perturbation-induced effects. The components of $\mathbf{z}_\nu$ are assumed to obey an unknown causal structure, constrained to be a directed acyclic graph (DAG).

Following standard causal modeling assumptions, we associate each latent causal variable $\mathbf{z}_i$ with an independent exogenous variable: $\mathbf{n}_\iota$ for $\mathbf{z}_\iota$ and $\mathbf{n}_{\nu,i}$ for each component of $\mathbf{z}_{\nu,i}$, capturing sources of information.

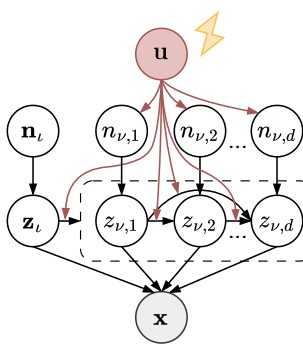

*Figure 2.* A latent generative model of single-cell perturbations. $\mathbf{z}_\iota$ denotes perturbation-invariant variable and $\mathbf{z}_\nu$ denotes perturbation-responsive variables.

Under the above generative assumptions, the joint prior distribution over the latent variables factorizes as

$$p(\mathbf{z}_\nu, \mathbf{z}_\iota \mid \mathbf{u}) = p(\mathbf{z}_\nu \mid \mathbf{u}, \mathbf{z}_\iota)\, p(\mathbf{z}_\iota), \tag{1}$$

where the perturbation-invariant variables $\mathbf{z}_\iota$ are independent of the perturbation condition $\mathbf{u}$, while the perturbation-responsive variables $\mathbf{z}_\nu$ depend on both the applied perturbation and the background cellular state.

### 3.2. Variational Inference Model

The posterior distribution over the latent variables, $p(\mathbf{z}_\nu, \mathbf{z}_\iota \mid \mathbf{x}, \mathbf{u})$, induced by the proposed generative model is generally intractable. We therefore adopt a variational inference to approximate the posterior. Specifically, we introduce the following structured variational distribution:

$$q(\mathbf{z}_\nu, \mathbf{z}_\iota \mid \mathbf{x}, \mathbf{u}) = q(\mathbf{z}_\nu \mid \mathbf{x}, \mathbf{u})\, q(\mathbf{z}_\iota \mid \mathbf{x}), \tag{2}$$

which reflects our modeling assumptions: $\mathbf{z}_\iota$ captures perturbation-invariant background information inferred from the observed expression $\mathbf{x}$, while $\mathbf{z}_\nu$ captures perturbation-responsive effects conditioned on both $\mathbf{x}$ and the perturbation condition $\mathbf{u}$.

Recall that the perturbation-responsive latent variables $\mathbf{z}_\nu$ are assumed to follow an underlying DAG, which captures the structured dependencies among perturbation effects. Accordingly, the inference model $q_\theta(\mathbf{z}_\nu \mid \mathbf{x}, \mathbf{u})$ in Eq. 2 is designed to recover both the latent variables and their dependency structure, so that the learned representations encode not only perturbation effects but also their organization. This structured representation is essential for generalization to unseen perturbations.

**Evidence Lower Bound.** Given the prior factorization in Eq. 1 and the variational posterior in Eq. 2, we optimize the conditional likelihood $\log p(\mathbf{x} \mid \mathbf{u})$ through the following evidence lower bound (ELBO):

$$\begin{aligned}
\mathcal{L}_{\text{ELBO}} &= \mathbb{E}_{q_\theta(\mathbf{z}_\nu, \mathbf{z}_\iota \mid \mathbf{x}, \mathbf{u})}\big[\log p(\mathbf{x} \mid \mathbf{z}_\nu, \mathbf{z}_\iota, \mathbf{u})\big] \\
&\quad - D_{\text{KL}}(q(\mathbf{z}_\nu, \mathbf{z}_\iota \mid \mathbf{x}, \mathbf{u}) \,\|\, p(\mathbf{z}_\nu, \mathbf{z}_\iota \mid \mathbf{u})). \quad (3)
\end{aligned}$$

**Contrastive Alignment.** Optimizing the ELBO alone may not resolve the *Perturbation Suppression Hypothesis*: when invariant background programs dominate gene expression data, reconstruction can be achieved primarily by modeling this dominant component. In this regime, the perturbation-responsive variables $\mathbf{z}_\nu$ may be under-recovered, as their contribution is overwhelmed by invariant background information, leading the learned representations to discard perturbation-induced signals.

To counteract this effect, we introduce a contrastive alignment objective that uses unperturbed controls as an explicit reference for background information. Specifically, for each perturbed sample $(\mathbf{x}, \mathbf{u})$, we sample a control expression profile $\mathbf{x}^{(\mathbf{u}_0)}$ from the unperturbed condition and encourage their *perturbation-invariant* representations to agree. Concretely, we align the invariant latents inferred from the two samples by minimizing

$$\mathcal{L}_{\text{contrast}}(\mathbf{x}, \mathbf{x}^{(\mathbf{u}_0)}) = \|\mathbf{z}_\iota - \mathbf{z}_\iota^{(\mathbf{u}_0)}\|_2^2, \tag{4}$$

where $\mathbf{z}_\iota \sim q_\theta(\mathbf{z}_\iota \mid \mathbf{x})$ and $\mathbf{z}_\iota^{(\mathbf{u}_0)} \sim q_\theta(\mathbf{z}_\iota \mid \mathbf{x}^{(\mathbf{u}_0)})$.

Intuitively, by enforcing consistency of $\mathbf{z}_\iota$ across perturbed and unperturbed samples, the alignment enforces dominant, perturbation-invariant variation to be explained through $\mathbf{z}_\iota$. Once this dominant background variation is accounted for in $\mathbf{z}_\iota$, it no longer needs to be explained by other latent variables during reconstruction. Consequently, the perturbation-responsive variables $\mathbf{z}_\nu$ are freed from modeling background structure and are instead driven to capture the residual, perturbation-induced changes.

### 3.3. Learning Objective

The final training objective combines the variational objective with the contrastive alignment term:

$$\mathcal{L} = -\mathcal{L}_{\text{ELBO}} + \alpha\, \mathcal{L}_{\text{contrast}}, \tag{5}$$

where $\alpha$ controls the strength of the alignment regularization. We refer to the resulting model as *Perturbation-aware Variational Autoencoders* (PerturbedVAE). Figure 3 provides an overview of the proposed framework.

**Objective Interpretation.** The objective in Eq. 5 balances several complementary goals. The reconstruction term in the ELBO ensures that the latent variables retain sufficient information to explain the observed gene expression. The KL regularization on $\mathbf{z}_\iota$ constrains the perturbation-invariant component, while the KL term on $\mathbf{z}_\nu$ encourages a compact representation of perturbation-responsive

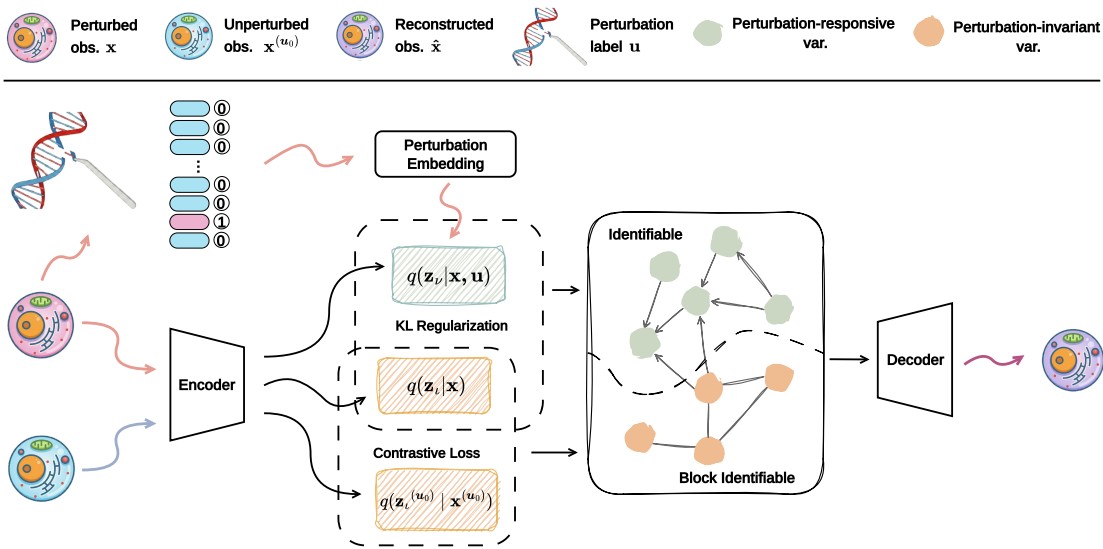

*Figure 3.* Framework of the proposed PerturbedVAE. Perturbed $\mathbf{x}$ are used to learn the perturbation-responsive block $\mathbf{z}_\nu$, which captures the effects of perturbations indexed by $\mathbf{u}$. In parallel, unperturbed $\mathbf{x}^{(\mathbf{u}_0)}$ are leveraged for contrastive alignment of the perturbation-invariant block $\mathbf{z}_\iota$, encouraging invariant background programs to be separated from perturbation-induced variation.

variation. The contrastive alignment term explicitly enforces the separation between perturbation-invariant and perturbation-responsive factors by anchoring background variation in $\mathbf{z}_\iota$. Together, these components enable PerturbedVAE to both extract perturbation-induced signals and organize them in a structured latent space that supports generalization to unseen perturbations.

**Test-time Prediction.** At test time, PerturbedVAE predicts a target perturbation without using post-perturbation expression as input. Given an unperturbed/control cell $\mathbf{x}^{(\mathbf{u}_0)}$ and a perturbation-condition vector $\mathbf{u}$, we infer the invariant latent variable $\mathbf{z}_\iota \sim q_\phi(\mathbf{z}_\iota \mid \mathbf{x}^{(\mathbf{u}_0)})$, generate the responsive latent variable $\mathbf{z}_\nu$ from the learned perturbation-conditioned mechanism $p_\theta(\mathbf{z}_\nu \mid \mathbf{z}_\iota, \mathbf{u})$, and decode $(\mathbf{z}_\iota, \mathbf{z}_\nu)$ to the predicted expression profile. For double-gene OOD prediction, we directly feed unseen two-hot perturbation vectors into the same learned $\mathbf{u}$-conditioned mechanism trained from single-gene interventions.

## 4. A Theoretically-Grounded Construction of the PerturbedVAE Framework

The framework introduced in Sec. 3 is deliberately kept implementation-agnostic and does not commit to specific parametric choices (e.g., the concrete form of the prior in Eq. 1). In this section, we provide a formal identifiability analysis that places the framework on a rigorous theoretical footing. By introducing explicit assumptions and parameterizations of the data-generating process, we establish theoretical guarantees for the proposed framework and,

in turn, clarify how it can be realized in practice while preserving identifiability guarantees.

Without additional assumptions, exactly recovering latent variables is in general impossible from the observed variables $\mathbf{x}$ and $\mathbf{u}$ alone, even in the simple nonlinear ICA setting (Hyvärinen & Pajunen, 1999; Hyvarinen & Morioka, 2016; 2017). To enable the theoretical analysis that follows, we therefore impose additional structure and parameterize the proposed causal generative model as follows.

$$\mathbf{z}_\iota := \boldsymbol{\lambda}_{\iota\iota}\,\mathbf{z}_\iota + \mathbf{n}_\iota, \qquad \mathbf{n}_\iota \sim \mathcal{N}\big(\boldsymbol{\mu}_\iota, \mathrm{diag}\,\boldsymbol{\beta}_\iota\big), \quad (6)$$

$$\mathbf{z}_\nu := \boldsymbol{\lambda}_{\nu\iota}(\mathbf{u})\,\mathbf{z}_\iota + \boldsymbol{\lambda}_{\nu\nu}(\mathbf{u})\,\mathbf{z}_\nu + \mathbf{n}_\nu, \quad (7)$$

$$\mathbf{n}_\nu \sim \mathcal{N}\big(\boldsymbol{\mu}_\nu(\mathbf{u}), \mathrm{diag}\,\boldsymbol{\beta}_\nu(\mathbf{u})\big), \quad (8)$$

$$\mathbf{x} := \mathrm{g}(\mathbf{z}), \quad (9)$$

where,

- $\mathbf{n}_\iota \in \mathbb{R}^{d_\iota}$ and $\mathbf{n}_\nu \in \mathbb{R}^{d_\nu}$ are latent noise variables, sampled from Gaussian, i.e., $\mathcal{N}\big(\boldsymbol{\mu}_\iota, \mathrm{diag}\,\boldsymbol{\beta}_\iota\big)$ and $\mathcal{N}\big(\boldsymbol{\mu}_\nu(\mathbf{u}), \mathrm{diag}\,\boldsymbol{\beta}_\nu(\mathbf{u})\big)$, respectively.

- The matrices $\boldsymbol{\lambda}_{\iota\iota}$, $\boldsymbol{\lambda}_{\nu\nu}(\mathbf{u})$, and $\boldsymbol{\lambda}_{\nu\iota}(\mathbf{u})$ are weight matrices and are assumed to be strictly lower triangular to satisfy the DAG constraint.[1] Here we assume linear causal relationships among the latent variables, primarily for simplicity and practical applicability, following common practice in prior work (Zhang et al., 2023; de la Fuente et al., 2025; Lopez et al., 2022).

---

[1]Without loss of generality, we fix a common acyclic ordering across environments, due to permutation indeterminacy in the latent space (Squires et al., 2023; Liu et al., 2026c).

- $\mathbf{z} = (\mathbf{z}_\iota, \mathbf{z}_\nu)$ and g denotes an unknown nonlinear mapping from $\mathbf{z}$ to $\mathbf{x}$.

Given the parameterization of the latent causal generative model above, we now introduce the following result.

**Theorem 1** (Identifiability Results). *Suppose the observed variable $\mathbf{x}$ and latent causal variables $\mathbf{z} = (\mathbf{z}_\iota, \mathbf{z}_\nu)$ follow the generative model defined in Eqs. 6–9, parameterized by $\boldsymbol{\theta} = (g, \boldsymbol{\lambda}, \boldsymbol{\mu}, \boldsymbol{\beta})$. Let $\hat{\boldsymbol{\theta}} = (\hat{g}, \hat{\boldsymbol{\lambda}}, \hat{\boldsymbol{\mu}}, \hat{\boldsymbol{\beta}})$ be the estimated parameters obtained by matching the conditional data distribution $p_\theta(\mathbf{x}|\mathbf{u}) = p_{\hat{\theta}}(\mathbf{x}|\mathbf{u})$ and minimizing the alignment loss. Assume the following conditions hold:*

*(i) **Invertibility and Smoothness:** The unknown nonlinear mapping g is smooth and invertible.*

*(ii) **Environmental Sufficiency:** There exist $2d_\nu$ distinct environments $\{\mathbf{u}_1, \ldots, \mathbf{u}_m\}$ relative to a reference $\mathbf{u}_0$ such that the matrix*

$$\mathbf{L}^\top = [\Delta\boldsymbol{\eta}(\mathbf{u}_1), \ldots, \Delta\boldsymbol{\eta}(\mathbf{u}_{2d_\nu})]^\top \in \mathbb{R}^{2d_\nu \times 2d_\nu} \quad (10)$$

*has full column rank $2d_\nu$, where (elementwise divisions)*

$$\Delta\eta(\mathbf{u}) := \begin{pmatrix} \frac{\boldsymbol{\mu}_\nu(\mathbf{u})}{\boldsymbol{\beta}_\nu(\mathbf{u})} - \frac{\boldsymbol{\mu}_\nu(\mathbf{u}_0)}{\boldsymbol{\beta}_\nu(\mathbf{u}_0)} \\ -\frac{1}{2}\left(\frac{1}{\boldsymbol{\beta}_\nu(\mathbf{u})} - \frac{1}{\boldsymbol{\beta}_\nu(\mathbf{u}_0)}\right) \end{pmatrix} \in \mathbb{R}^{2d_\nu}. \quad (11)$$

*(iii) **Optimal Alignment:** The alignment loss, e.g., Eq. 4 attains its global minimum such that $\mathbf{f}_\iota(\mathbf{x}^{(\mathbf{u})}) = \mathbf{f}_\iota(\mathbf{x}^{(\mathbf{u}_0)})$ almost surely for any $\mathbf{u}, \mathbf{u}_0$, where $\mathbf{f}_\iota = \hat{g}_\iota^{-1}$.*

*(iv) **Intervention Sufficiency:** The function class of $\boldsymbol{\lambda}$ satisfies the following condition: there exists $\mathbf{u}_i$, such that, for all parent nodes $z_j \in \mathrm{pa}_i$ of $z_i$, $\boldsymbol{\lambda}_{j,i} = 0$.*

*Then, the true latent causal variables $\mathbf{z}$ are related to the variables $\hat{\mathbf{z}}$ estimated by matching likelihood (via the ELBO objective in Eq. 3) as follows:*

*1. $\mathbf{z}_\nu$ is identified up to permutation and scaling, i.e., $\mathbf{z}_\nu = \mathbf{P}_\nu\hat{\mathbf{z}}_\nu + \mathbf{c}_\nu$, here $\mathbf{P}_\nu$ is a permutation matrix with scaling.*

*2. $\mathbf{z}_\iota$ is identified up to a linear block transformation, i.e., $\mathbf{z}_\iota = \mathbf{A}_\iota\hat{\mathbf{z}}_\iota + \mathbf{c}_\iota$, where $\mathbf{A}_\iota$ is a non-singular matrix.*

*Proof.* Proof can be found in App. F. □

Refer to Sec. E for a justification of the assumptions.

**Discussion.** The identifiability result in Theorem 1 characterizes the conditions under which perturbation-specific causal variables can be reliably recovered from partial-intervention data. Importantly, these conditions directly inform how the PerturbedVAE framework is instantiated.

**Contrastive Loss in Eq. 5.** Theorem 1 highlights the importance of preventing perturbation-induced variation from being absorbed into invariant representations, by assumption (iii). However, this requirement is often overlooked in prior causal representation learning approaches (Zhang et al., 2023; de la Fuente et al., 2025; Lopez et al., 2022). As a result, these methods may struggle in practice under partial-intervention settings. Further, assumption (iii) is aligned with the *contrastive alignment* objective, which explicitly enforces Assumption (iii) by encouraging the inferred invariant representation $\hat{\mathbf{z}}_\iota = \mathbf{f}_\iota(\mathbf{x})$ to remain identical across perturbed and unperturbed conditions. As a result, environment-dependent information cannot be explained away by $\mathbf{z}_\iota$ and is instead forced to be captured by the perturbation-responsive variables $\mathbf{z}_\nu$.

**ELBO Loss in Eq. 5.** Theorem 1 not only clarifies the role of environmental diversity in achieving identifiability, but also directly constrains how the latent variable model should be parameterized, thereby guiding the concrete construction of the ELBO objective in Eq. 5. Specifically, under the proposed latent causal generative model (Eqs. 6-9), both the prior and posterior of the perturbation-responsive variables $\mathbf{z}_\nu$ are parameterized as linear Gaussian structural causal models conditioned on the perturbation label $\mathbf{u}$. This environment-dependent parameterization is required by the identifiability conditions and is explicitly reflected in the ELBO through perturbation-conditioned distributional parameters. In contrast, the invariant variables $\mathbf{z}_\iota$ are identifiable only up to a linear block transformation. Accordingly, we model $\mathbf{z}_\iota$ using an i.i.d. Gaussian form and omit additional structural constraints for simplicity. See Sec. G for a specific implementation of the proposed PerturbedVAE.

## 5. Empirical Findings

### 5.1. Numerical Simulation

We first conduct simulations to verify our theoretical results under controllable assumptions. To this end, we generate synthetic data according to our latent causal generative model in Eqs. 6- 9. More details can be found in App. H.1. This setup allows us to systematically assess the recovery of the latent subspace over $\mathbf{z}_\iota$, and causal structure recovery over latent perturbed variables $\mathbf{z}_\nu$. For evaluation, following Sorrenson et al. (2020); Khemakhem et al. (2020), we use the mean correlation coefficient (MCC) to quantify component-wise recovery of $\mathbf{z}_\nu$. Specifically, MCC measures the correlation between each learned component of $\mathbf{z}_\nu$ and its corresponding ground-truth component, with a value of 1 indicating perfect recovery. For block identifi-

*Table 1.* Results on simulated data. MCC evaluates component-wise recovery of $\mathbf{z}_\nu$, and $R^2$ evaluates block-level recovery of $\mathbf{z}_\iota$. Contrastive alignment consistently improves identifiability and disentanglement.

| Contrastive Alignment | MCC | $R^2$ | |
|---|---|---|---|
| | Var. $\mathbf{z}_\nu$ (identifiable) | Var. $\mathbf{z}_\nu$ (block-identifiable) | Inv. $\mathbf{z}_\iota$ (block-identifiable) |
| ✗ | $0.81_{\pm 0.0306}$ | $0.93_{\pm 0.0120}$ | $0.66_{\pm 0.0281}$ |
| ✓ | $\mathbf{0.86}_{\pm 0.0285}$ | $\mathbf{0.95}_{\pm 0.0020}$ | $\mathbf{0.97}_{\pm 0.0077}$ |

ability evaluation of $\mathbf{z}_\iota$, we report the regression $R^2$, following Von Kügelgen et al. (2021), which measures the correlation between the learned block and its ground-truth counterpart. Values closer to 1 are better.

Table 1 shows that the contrastive alignment term substantially improves identifiability. For the variant block $\mathbf{z}_\nu$, MCC increases from 0.81 to 0.86 and block-wise $R^2$ from 0.93 to 0.95, indicating more accurate recovery of intervention-specific factors. The effect is even more pronounced for the invariant block $\mathbf{z}_\iota$, whose $R^2$ rises from 0.66 to 0.97, highlighting the crucial role of contrastive alignment in disentangling invariant programs from perturbation-induced effects. These results confirm our theoretical claims as stated in Theorem 1.

### 5.2. Experiments on Real Data

For real-world perturbation data, we consider the large-scale Perturb-seq dataset from (Norman et al., 2019) (See App. H.2 for more details), following previous works (Zhang et al., 2023; de la Fuente et al., 2025; Lopez et al., 2022; Bereket & Karaletsos, 2023). We design two types of comparisons. First, we compare our method with FMs to evaluate whether the proposed method provides advantages over large pretrained models. Second, we compare with existing methods based on causal representation learning, to assess the benefits of our contrastive and causal modeling components beyond causal modeling alone.

*Table 2.* Double-gene perturbation prediction results. RMSE ($\downarrow$) and $R^2$ ($\uparrow$) demonstrate that our method achieves improved generalization to combinatorial perturbations compared to others.

| Method | Metrics | |
|---|---|---|
| | RMSE | $R^2$ |
| **scFoundation** (Hao et al., 2024) | $0.5714_{\pm 0.0105}$ | $0.9844_{\pm 0.006}$ |
| **UCE** (Rosen et al., 2023) | $0.5634_{\pm 0.0039}$ | $\underline{0.9857_{\pm 0.0006}}$ |
| **Geneformer** (Theodoris et al., 2023) | $0.6132_{\pm 0.0322}$ | $0.9728_{\pm 0.0015}$ |
| **STATE** (Adduri et al., 2025) | $0.4981_{\pm 0.0046}$ | $0.9475_{\pm 0.0021}$ |
| **RandomForest** (Breiman, 2001) | $0.4931_{\pm 0.0003}$ | $0.9800_{\pm 0.0005}$ |
| **ElasticNet** (Zou & Hastie, 2005) | $0.4929_{\pm 0.0000}$ | $0.9795_{\pm 0.0000}$ |
| **KNN** (Cover & Hart, 1967) | $0.4894_{\pm 0.0000}$ | $0.9843_{\pm 0.000}$ |
| **PerturbedVAE (Ours)** | $\mathbf{0.4474}_{\pm 0.0007}$ | $\mathbf{0.9865}_{\pm 0.0009}$ |

**Compared With FMs.** We benchmark PerturbedVAE against widely used single-cell FMs, including scFoundation (Hao et al., 2024), UCE (Rosen et al., 2023), and Geneformer (Theodoris et al., 2023), STATE (Adduri et al., 2025). For each model, we follow the recommended fine-tuning and inference protocol. All methods are evaluated under the same data splits and metrics. Under cell-wise evaluation, Table 2 shows that FMs degrade substantially in the setting of double-gene perturbations, whereas PerturbedVAE achieves consistently better performance across five random seeds. Together with our diagnostic analysis in Sec. 2 (Figure 1), these results support that FM representations retain limited perturbation-specific information, which in turn constrains their ability to support accurate perturbation prediction. In contrast, the proposed method explicitly preserves perturbation-specific information, and more importantly, effectively utilizes it by causal modeling, leading to improved performance.

**Compared With Simple Baselines.** We also compared against simple baselines, including RandomForest (Breiman, 2001), ElasticNet (Zou & Hastie, 2005), and KNN (Cover & Hart, 1967). Recent benchmarking studies have shown that, in certain settings, FMs do not consistently outperform such simple baselines (Csendes et al., 2025). In contrast, our proposed method achieves superior performance compared to these baselines, as shown in Table 2, suggesting that incorporating inductive biases and learning perturbation-aware representations may lead to improved generalization.

**Perspective on the Additive Linear Baseline.** Recent work by Ahlmann-Eltze et al. (2025) reports that a classical additive linear model can achieve surprisingly strong performance in combinatorial perturbation settings. We therefore compare PerturbedVAE with the additive model, with full results reported in App. M. Consistent with prior observations (Ahlmann-Eltze et al., 2025), the additive model performs competitively on this dataset, suggesting that many combinatorial perturbation effects are approximately linear and exhibit weak interactions.

At the cell level, however, the additive model exhibits a different trade-off. As shown in Table 3, it attains a slightly lower RMSE on genome-wide expression profiles, but its cell-level $R^2$ is negative and therefore not informative. In contrast, PerturbedVAE achieves a valid and high cell-level $R^2$, suggesting better preservation of cell-level explained variance under double-gene OOD prediction.

Nevertheless, such linear compositionality is not guaranteed to hold in general, particularly in the presence of nonlinear gene–gene interactions or context-dependent effects. Our analysis therefore clarifies both the strengths and limitations of simple additive models, and positions PerturbedVAE as a structured alternative that remains applicable when linearity assumptions break down. For completeness, we also report supplementary PCA/scVI-based

*Table 3.* Comparison with the additive linear baseline on genome-wide expression profiles under double-gene perturbation prediction results. A dash (–) indicates negative $R^2$.

| Method | RMSE $\downarrow$ | $R^2 \uparrow$ |
|---|---|---|
| Additive Linear | $\mathbf{0.4424}_{\pm \mathbf{0.0000}}$ | – |
| PerturbedVAE | $0.4494_{\pm \mathbf{0.0008}}$ | $\mathbf{0.9840}_{\pm \mathbf{0.0011}}$ |

additive controls in App. N.

### 5.3. Compared with Existing Latent Causal Models

**Experimental Setup.** To compare with existing methods that learn causal representations, we follow the experimental protocol of Zhang et al. (2023) (See App. H.2 for more details). We benchmark PerturbedVAE against four representative baselines, Discrepancy-VAE (Zhang et al., 2023), SENA-discrepancy-VAE (SENA) (de la Fuente et al., 2025), sVAE+ (Lopez et al., 2022), SAMS-VAE (Bereket & Karaletsos, 2023), reporting results averaged over five random seeds for each model. We also implement a variant of the proposed PerturbedVAE, namely PerturbedVAE (w/o Align), which excludes the contrastive term.

**Single-Gene Perturbation.** We first assess whether the compared latent causal representation models can reproduce observed single-gene perturbation responses. On Norman2019, we evaluate generative fidelity using the 14 single-gene conditions with more than 800 cells. For each condition, we generate 96 synthetic cells from the trained model and compare them to 96 held-out real cells not used during training. Performance is evaluated using population-level $R^2$ and RMSE across all genes (Table 4). Our model achieves consistently high fidelity, with an average $R^2$ of 0.99 across the 14 conditions (Left in Figure 4), indicating that it reproduces the mean perturbation response.

We further evaluate the latent causal model comparison on Replogle et al. (2022) as an additional cross-dataset single-gene i.i.d. check. PerturbedVAE again shows the strongest representation-learning performance, with full results in App. K.

**Double-Gene Perturbation.** We further evaluate 112 double-gene perturbations, which constitute a zero-shot prediction setting as no combinatorial conditions are observed during training. For each perturbation, we compare the population-average expression profile of generated cells with that of the held-out real cells. Despite this challenge, PerturbedVAE achieves strong performance, with an average $R^2$ of 0.98 across all genes (right panel of Figure 4), indicating that it successfully composes knowledge from single-gene interventions to predict unseen combinatorial effects. Consistently low RMSE values in Table 4 further show that the model preserves absolute expression magnitudes under out-of-distribution conditions.

Taken together, these results demonstrate that PerturbedVAE not only outperforms existing methods under observed single-gene perturbations, but also generalizes reliably to unseen combinatorial perturbations, highlighting the benefit of explicitly isolating perturbation-specific signals and aligning with our theoretical analysis.

### 5.4. Representation Diagnostics and Interpretability

**Unperturbed Latent Subspace.** For the invariant block $\mathbf{z}_\iota$, we examine whether its representation remains stable across perturbations, as a qualitative check of the intended block-level invariance property. Specifically, we analyze all perturbation conditions in the test set and visualize the inferred $\mathbf{z}_\iota$ representations, which are provided in App. I.

**Perturbed Latent Subspace: Biological Plausibility Check.** Complementary to the unperturbed latent subspace analysis above, we perform an analysis of perturbed latent space; we first provide the learned causal graph by PerturbedVAE to assess whether it captures biologically meaningful regulatory patterns. Following Zhang et al. (2023), we derive a program-level directed acyclic graph by assigning each latent program to a target gene using a simple heuristic based on maximal intervention effect. The resulting graph (Figure 5) recovers several well-established regulatory interactions, including the TGFBR2→SNAI1 axis involved in epithelial–mesenchymal transition (EMT) (Vincent et al., 2009; Fan et al., 2025), the canonical TP73→CDKN1A tumor suppressor pathway governing cell-cycle arrest (Schmidt et al., 2021), and the inhibitory regulation of JUN by DUSP9 in MAPK signaling (Emanuelli et al., 2008). While this analysis is not intended as a comprehensive biological validation, the recovery of these known mechanisms provides supportive evidence that the learned latent structure is not arbitrary and reflects meaningful aspects of underlying regulatory processes, consistent with the goal of learning structured and causally interpretable representations. More details and further analysis can be found in App. J.1.

**Perturbed Latent Subspace: Information Preservation.**

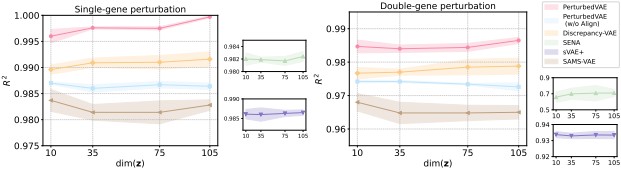

*Figure 4.* $R^2$ scores for genetic perturbation prediction across different latent dimensions $\dim(\mathbf{z})$. **Left:** single-gene perturbations. **Right:** double-gene perturbations. The proposed method consistently achieves higher $R^2$ and remains stable as $\dim(\mathbf{z})$ increases. Shaded areas denote variation across runs.

*Table 4.* RMSE (lower is better) for single- and double-gene perturbation prediction across different numbers of perturbed genes. Mean ± standard deviation over multiple runs are reported. Compared to existing latent causal representation methods, PerturbedVAE yields consistently lower errors, and the alignment mechanism further improves generalization to unseen double-gene perturbations.

| Method | Single-gene | | | | Double-gene | | | |
|---|---|---|---|---|---|---|---|---|
| | 10 | 35 | 75 | 105 | 10 | 35 | 75 | 105 |
| Discrepancy-VAE (Zhang et al., 2023) | $0.5603_{\pm 0.0030}$ | $0.5560_{\pm 0.0027}$ | $0.5582_{\pm 0.0038}$ | $0.5558_{\pm 0.0022}$ | $0.6084_{\pm 0.0045}$ | $0.6037_{\pm 0.0025}$ | $0.6075_{\pm 0.0072}$ | $0.6082_{\pm 0.0045}$ |
| SENA (de la Fuente et al., 2025) | $0.5839_{\pm 0.0021}$ | $0.5837_{\pm 0.0086}$ | $0.5778_{\pm 0.0109}$ | $0.5837_{\pm 0.0074}$ | $0.8573_{\pm 0.0205}$ | $0.8514_{\pm 0.0248}$ | $0.8507_{\pm 0.0396}$ | $0.8483_{\pm 0.0248}$ |
| sVAE+ (Lopez et al., 2022) | $0.5012_{\pm 0.0018}$ | $0.5005_{\pm 0.0025}$ | $0.5003_{\pm 0.0024}$ | $0.5002_{\pm 0.0022}$ | $0.5663_{\pm 0.0009}$ | $0.5667_{\pm 0.0008}$ | $0.5665_{\pm 0.0011}$ | $0.5664_{\pm 0.0012}$ |
| SAMS-VAE (Bereket & Karaletsos, 2023) | $0.4114_{\pm 0.0020}$ | $0.4136_{\pm 0.0019}$ | $0.4140_{\pm 0.0022}$ | $\underline{0.4123_{\pm 0.0029}}$ | $0.4605_{\pm 0.0020}$ | $0.4631_{\pm 0.0024}$ | $0.4632_{\pm 0.0017}$ | $0.4629_{\pm 0.0014}$ |
| **PerturbedVAE (w/o Align) (ours)** | $0.4098_{\pm 0.0001}$ | $0.4115_{\pm 0.0008}$ | $0.4115_{\pm 0.0005}$ | $0.4155_{\pm 0.0038}$ | $\underline{0.4557_{\pm 0.0005}}$ | $\underline{0.4563_{\pm 0.0005}}$ | $\underline{0.4577_{\pm 0.0005}}$ | $\underline{0.4623_{\pm 0.0041}}$ |
| **PerturbedVAE (ours)** | $\mathbf{0.4027_{\pm 0.0028}}$ | $\mathbf{0.3998_{\pm 0.0013}}$ | $\mathbf{0.3997_{\pm 0.0013}}$ | $\mathbf{0.3995_{\pm 0.0013}}$ | $\mathbf{0.4493_{\pm 0.0019}}$ | $\mathbf{0.4494_{\pm 0.0008}}$ | $\mathbf{0.4489_{\pm 0.0009}}$ | $\mathbf{0.4474_{\pm 0.0007}}$ |

We further evaluate whether perturbation-specific information is preserved by analyzing performance on the top 20 differentially expressed (DE) genes under perturbation, which form a perturbation-enriched 20-dimensional subspace. This serves as a diagnostic probe of whether perturbation-induced variation is retained rather than suppressed by invariant background programs. Results are reported in App. J.2.

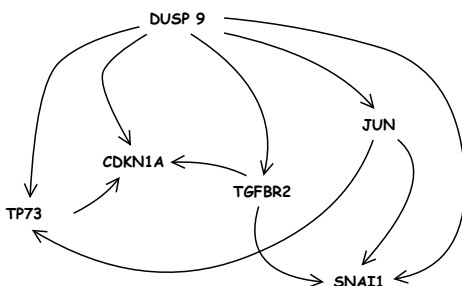

*Figure 5.* Learned causal structure over the perturbation-responsive latent variables $\mathbf{z}_\nu$. Nodes correspond to latent programs assigned to target genes via maximal intervention effect, and directed edges indicate inferred causal dependencies. Several known regulatory relationships are recovered.

## 6. Conclusion

We showed that the failure of foundation models in perturbation prediction is not primarily a matter of scale, but stems from a mismatch between generic training objectives and the sparse, intervention-driven structure of perturbation data. Motivated by this, we proposed PerturbedVAE, which introduces explicit inductive biases to disentangle invariant and perturbation-responsive factors and to organize the latter via a latent causal structure. We provided theoretical identifiability results and empirical evidence on both synthetic and real datasets demonstrating accurate prediction and interpretable structure. Our results suggest that a good representation for single-cell perturbation prediction is one that is perturbation-aware and causally structured, enabling reliable generalization under interventions.

## Acknowledgment

The work is partly supported by the Responsible AI Research Centre and ARC Discovery Project DP240103278.

## Impact Statement

This paper presents work whose goal is to advance the field of machine learning. There are many potential societal consequences of our work, none of which we feel must be specifically highlighted here.

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

# Part I

# Appendix

## Table of Contents

# A. Related Work

**Disentangling Single-Cell Perturbation Effects.** A central challenge in single-cell perturbation modeling is to separate intervention effects from intrinsic cellular variability. Deep generative approaches have shown strong performance on this task. scGen (Lotfollahi et al., 2019) models perturbations as additive shifts in a latent space, while CPA (Lotfollahi et al., 2023) factorizes each cell into basal state and perturbation effect. chemCPA (Hetzel et al., 2022) extends CPA with chemical structure embeddings and dosage information, enabling zero-shot predictions for unseen compounds. Other methods incorporate biological priors or contrastive objectives: GEARS (Roohani et al., 2024) uses gene-gene interaction graphs for improved generalization across perturbation combinations, and contrastive VAEs have been applied in optical pooled screening to disentangle stable identity from perturbation-driven variation (Wang et al., 2023). Despite empirical successes, most of these models treat disentanglement statistically rather than causally, which limits interpretability. Recent work has incorporated sparsity into latent-variable models to encourage identifiable and interpretable representations. CausCell (Gao et al., 2025b) enables counterfactual generation via SCM-guided diffusion, but critically depends on a predefined causal graph, limiting its applicability when causal structures are unknown or hard to specify. sVAE+ (Lopez et al., 2022), SAMS-VAE (Bereket & Karaletsos, 2023) impose sparse structure or mechanism shifts in the latent space to model perturbation-induced variation. Recent advances such as discrepancy-VAE (Zhang et al., 2023), and its interpretable variant (de la Fuente et al., 2025) align latent-variable models with identifiable causal semantics, pointing toward representations that are both intervention-sensitive and explanatory. Building on these advances, our approach moves beyond purely statistical factorization, ensuring that the learned representations reflect genuine causal effects of perturbations.

**Identifiable Causal Representations.** A key aim in modeling complex systems is to learn low-dimensional latent variables $\mathbf{z}$ from high-dimensional data $\mathbf{x}$ that match the true generative factors (Hyvärinen et al., 2001; Liu et al., 2026a). Nonlinear ICA showed that such components are not identifiable from i.i.d. data without extra assumptions (Hyvärinen & Pajunen, 1999). Identifiable variants address this by introducing an auxiliary variable $\mathbf{u}$ so that latent factors $\{z_i\}_{i=1}^p$ are conditionally independent given $\mathbf{u}$ (Hyvarinen & Morioka, 2016; 2017). The iVAE framework (Khemakhem et al., 2020), built on VAEs (Kingma et al., 2013; Rezende et al., 2014), proves identifiability of both $\mathbf{z}$ and $p(\mathbf{x} \mid \mathbf{z})$ under mild conditions. Recent approaches impose structure in latent space: DAG-based models enforce acyclicity (Yang et al., 2021; Lippe et al., 2022b; Liu et al., 2026c; 2024b; Ahuja et al., 2023; Liu et al., 2024a; 2025), while factorized designs split latent variables into invariant, intervention-specific, and interaction parts (Von Kügelgen et al., 2021; Kong et al., 2022; Gao et al., 2025a). While prior methods establish identifiability via auxiliary conditioning or broad structural constraints, our model ties perturbations directly to latent mechanisms. This design moves beyond heuristic augmentations or globally factorized latents, making our framework specifically tailored to single-cell perturbation.

**Contrastive Representation Learning.** Contrastive multi-view learning learns invariances across views or modalities (e.g., SimCLR, BYOL, CLIP-style training) but typically relies on heuristic augmentations whose invariants need not align with causal structure (Chen et al., 2020; Grill et al., 2020; Radford et al., 2021; Cai et al., 2024; 2025; Liu et al., 2026b; Tschannen et al., 2020; Von Kügelgen et al., 2021). Aliee et al. (2023) learn conditionally invariant representations by leveraging variability across observational environments (patients, batches, platforms) to suppress domain-specific artifacts while preserving biological signal. In single-cell analysis, Weinberger et al. (2023) contrast background and target datasets—extending to multi-omics—to isolate salient structure, but provide no identifiability guarantees. For perturbation screens, supervised contrastive VAEs use guide labels with HSIC to isolate perturbation effects from background heterogeneity (Tu et al., 2024). Concurrently, Mao et al. (2024) posit a three-way factorization (covariate, treatment, interaction) and promote independence via structural constraints and adversarial training; while principled, this fixed design may underfit non-classical responses, and its identifiability hinges on stringent experimental designs. Unlike contrastive or domain-invariant models, we obtain block identifiability for the perturbation-invariant block and component-wise identifiability for the perturbation-responsive block under a weight-variant latent SCM, thereby performing CRL in the latent space and recovering the latent causal graph among responsive variables.

## B. Limitations

A fundamental limitation of this work, shared with most studies on latent causal representation learning, lies in the assumptions introduced in our theoretical analysis. Nevertheless, identifying latent causal variables without assumptions is impossible in general. As a result, like most identifiability analyses in the causal representation learning literature, we necessarily impose a set of assumptions to characterize when latent causal representations can be recovered. These assumptions are not specific to our framework; rather, they are commonly used in nonlinear ICA and causal representation learning, as discussed in Sec. E.

We further acknowledge that these assumptions are generally untestable in real-world biological data, similar to most of the work in the causal representation learning community, since the true generative process and latent causal variables are unknown. Nevertheless, our empirical results on real perturbation datasets suggest that our approach is effective, demonstrating that the insights derived from our theoretical analysis may meaningfully guide practical modeling.

Finally, our OOD claim is restricted to combinatorial generalization over genes observed during training. Specifically, the model is trained on single-gene perturbations and evaluated on unseen double-gene combinations, where each individual gene has already appeared as a perturbation target. This differs from unseen-gene generalization, where the model must predict the effect of perturbing a gene that was never observed during training. Our current perturbation input $\mathbf{u}$ is an identifier-style condition vector; it provides no biological geometry between genes and no information about the function, pathway membership, regulatory context, or sequence properties of an unseen target. Consequently, a new gene coordinate in $\mathbf{u}$ has no learned support, making unseen-gene prediction statistically underdetermined without additional side information. Addressing this setting would require augmenting $\mathbf{u}$ with biologically meaningful gene-level features, such as GO/pathway annotations, gene embeddings, regulatory-network features, or sequence-derived representations. We therefore do not claim to address unseen-gene or cross-cell-type perturbation generalization in this work.

## C. Difference from Existing Causal Representation Learning Methods

Learning causal representations for single-cell perturbation prediction is a promising direction. A central challenge in this line of work lies in theoretical support from identifiability analysis, which ensures that the true latent causal variables can be reliably recovered, thereby enabling meaningful learning of the causal relationships among them (Lachapelle et al., 2022; Zhang et al., 2023; Lopez et al., 2022; de la Fuente et al., 2025). Most existing identifiability results, however, rely on the assumption that *all* latent causal variables are intervened across environments (Zhang et al., 2023; Lopez et al., 2022; de la Fuente et al., 2025). In real cellular perturbation experiments, this assumption is rarely satisfied. Comprehensive perturbation of all genes is often prohibitively expensive or technically infeasible, and practical datasets typically contain interventions on only a small subset of genes, as we highlighted throughout this work. As a consequence, a large subspace of genes, and the corresponding latent causal variables governing their expression, remains unperturbed across all observed conditions.

This mismatch between theoretical assumptions and experimental reality poses a fundamental challenge. As a result, existing identifiability theory may not be directly applicable under such partial-intervention settings. In turn, methods built upon these theoretical results (Lopez et al., 2022; Zhang et al., 2023; de la Fuente et al., 2025) may struggle to perform effectively in practice, as invariant background variation can dominate the learned representations and obscure perturbation-specific signals.

In contrast to prior work, our approach explicitly embraces the partial-intervention nature of real cellular data. Rather than assuming that all latent causal variables are perturbed, we focus on separating perturbation-responsive factors from dominant perturbation-invariant structure. This perspective motivates our theoretical analysis, which, in turn, provides model design for the realistic setting of limited and partial perturbations, bridging the gap between identifiability theory and real-world single-cell perturbation data.

## D. Lemmas

**Lemma D.1** (Unit-Jacobian mapping from latent causal variables to latent noise variables)**.** *Consider the structural Eqs. 6-8. Let* $\mathbf{z} = (\mathbf{z}_\iota, \mathbf{z}_\nu) \in \mathbb{R}^{d_\iota + d_\nu}$ *and* $\mathbf{n} = (\mathbf{n}_\iota, \mathbf{n}_\nu) \in \mathbb{R}^{d_\iota + d_\nu}$*. Define the block matrix*

$$\mathbf{\Lambda}(\mathbf{u}) := \begin{pmatrix} \boldsymbol{\lambda}_{\iota\iota} & \mathbf{0} \\ \boldsymbol{\lambda}_{\nu\iota}(\mathbf{u}) & \boldsymbol{\lambda}_{\nu\nu}(\mathbf{u}) \end{pmatrix}. \tag{12}$$

*Then we have:*

$$\mathbf{z} = \mathbf{\Lambda}(\mathbf{u})\,\mathbf{z} + \mathbf{n}, \qquad \textit{equivalently} \qquad \mathbf{n} = (\mathbf{I} - \mathbf{\Lambda}(\mathbf{u}))\,\mathbf{z}. \tag{13}$$

*Moreover, the mapping* $\mathbf{z} \mapsto \mathbf{n}$ *is bijective and has unit Jacobian determinant:*

$$\big| \det(\mathbf{I} - \mathbf{\Lambda}(\mathbf{u})) \big| = 1 \qquad \textit{for all } \mathbf{u}. \tag{14}$$

*Proof.* By DAG assumption, $\boldsymbol{\lambda}_{\iota\iota}$ and $\boldsymbol{\lambda}_{\nu\nu}(\mathbf{u})$ are strictly lower triangular, hence have zeros on their diagonals. Therefore $\mathbf{\Lambda}(\mathbf{u})$ has zeros on its diagonal as well, and $I - \mathbf{\Lambda}(\mathbf{u})$ is block lower triangular with diagonal blocks $\mathbf{I} - \boldsymbol{\lambda}_{\iota\iota}$ and $\mathbf{I} - \boldsymbol{\lambda}_{\nu\nu}(\mathbf{u})$. Since each diagonal block is lower triangular with ones on the diagonal, we have

$$\det(\mathbf{I} - \boldsymbol{\lambda}_{\iota\iota}) = 1, \qquad \det(\mathbf{I} - \boldsymbol{\lambda}_{\nu\nu}(\mathbf{u})) = 1. \tag{15}$$

Hence

$$\det(\mathbf{I} - \mathbf{\Lambda}(\mathbf{u})) = \det(\mathbf{I} - \boldsymbol{\lambda}_{\iota\iota}) \cdot \det(\mathbf{I} - \boldsymbol{\lambda}_{\nu\nu}(\mathbf{u})) = 1, \tag{16}$$

which proves Eq. 14. In particular, $\mathbf{I} - \mathbf{\Lambda}(\mathbf{u})$ is invertible, so $\mathbf{z} \mapsto \mathbf{n}$ is bijective. $\qquad\square$

**Lemma D.2** (Component-wise identifiability of $\mathbf{n}_\nu$)**.** *Under the proposed latent generative model Eqs. 6-9 with an invertible g. Suppose there exist environments* $\{\mathbf{u}_0, \ldots, \mathbf{u}_{2d_\nu}\}$ *such that the matrix*

$$\mathbf{L}^\top := \begin{bmatrix} \Delta\eta(\mathbf{u}_1)^\top \\ \vdots \\ \Delta\eta(\mathbf{u}_{2d_\nu})^\top \end{bmatrix} \in \mathbb{R}^{2d_\nu \times 2d_\nu} \quad \textit{has full column rank } 2d_\nu, \tag{17}$$

*where (elementwise divisions)*

$$\Delta\boldsymbol{\eta}(\mathbf{u}) := \begin{pmatrix} \frac{\boldsymbol{\mu}_\nu(\mathbf{u})}{\boldsymbol{\beta}_\nu(\mathbf{u})} - \frac{\boldsymbol{\mu}_\nu(\mathbf{u}_0)}{\boldsymbol{\beta}_\nu(\mathbf{u}_0)} \\ -\frac{1}{2}\left( \frac{1}{\boldsymbol{\beta}_\nu(\mathbf{u})} - \frac{1}{\boldsymbol{\beta}_\nu(\mathbf{u}_0)} \right) \end{pmatrix} \in \mathbb{R}^{2d_\nu}. \tag{18}$$

*If two parameter sets* $\boldsymbol{\theta}, \hat{\boldsymbol{\theta}}$ *induce the same* $p(\mathbf{x} \mid \mathbf{u})$ *for all* $(\mathbf{x}, \mathbf{u})$*, then there exist a permutation* $\pi$ *and nonzero scalars* $\{a_j\}$ *and constants* $\{b_j\}$ *such that*

$$n_{\nu,j} = a_j\, \hat{n}_{\nu,\pi(j)} + b_j, \qquad j = 1, \ldots, d_\nu, \tag{19}$$

*where* $\hat{n}_{\nu,\pi(j)}$ *is estimated by the likelihood matching.*

**Proof Sketch.** Let $\boldsymbol{\theta} := \big(\text{g}, \boldsymbol{\lambda}_{\iota\iota}, \boldsymbol{\lambda}_{\nu\iota}(\cdot), \boldsymbol{\lambda}_{\nu\nu}(\cdot), \boldsymbol{\mu}_\iota, \boldsymbol{\beta}_\iota, \boldsymbol{\mu}_\nu(\cdot), \boldsymbol{\beta}_\nu(\cdot)\big)$ denote the collection of all model parameters, and let $\hat{\theta}$ denote another parameter set from the same model class. *(Step 1)* Using the change-of-variables formula and the unit Jacobian of the structural map $\mathbf{z} \mapsto \mathbf{n}$, equality $p_{\boldsymbol{\theta}}(\mathbf{x} \mid \mathbf{u}) \equiv p_{\hat{\boldsymbol{\theta}}}(\mathbf{x} \mid \mathbf{u})$ implies equality of $\log p_{\boldsymbol{\theta}}(\mathbf{n} \mid \mathbf{u})$ and $\log p_{\hat{\boldsymbol{\theta}}}(\hat{\mathbf{n}} \mid \mathbf{u})$ up to an $\mathbf{u}$-independent term, which vanishes upon differencing across environments; since $p_{\boldsymbol{\theta}}(\mathbf{n} \mid \mathbf{u}) = p(\mathbf{n}_\iota)\, p(\mathbf{n}_\nu \mid \mathbf{u})$ with $p(\mathbf{n}_\iota)$ invariant, this leaves an identity involving only $\mathbf{n}_\nu$. *(Step 2)* For diagonal Gaussians, the environment log-ratio is linear in the sufficient statistics $\mathbf{T}_\nu(\mathbf{n}_\nu) = (\mathbf{n}_\nu, \mathbf{n}_\nu^{\odot 2})$, and stacking over sufficiently many environments yields an affine relation $\mathbf{T}_\nu(\mathbf{n}_\nu) = \mathbf{A}_\nu \hat{\mathbf{T}}(\hat{\mathbf{n}}) + \mathbf{c}_\nu$ with $\mathbf{A}_\nu$ full rank. *(Step 3)* The affine relation implies each $n_{\nu,j}$ (and $n_{\nu,j}^2$) is a low-degree polynomial in $\hat{\mathbf{n}}$, and independence and Gaussianity then force each $n_{\nu,j}$ to depend on exactly one coordinate of $\hat{\mathbf{n}}$, implying identifiability up to permutation and component-wise scaling/shift.

*Proof.* **Step 1.** By Lemma D.1, for any $\mathbf{u}$ the map $\mathbf{z} \mapsto \mathbf{n} = (\mathbf{I} - \mathbf{\Lambda}(\mathbf{u}))\mathbf{z}$ is bijective and satisfies $\big| \det(\mathbf{I} - \mathbf{\Lambda}(\mathbf{u})) \big| = 1$. Since $\mathbf{x} = \text{g}(\mathbf{z})$ with g invertible, we can write

$$\mathbf{n} = (\mathbf{I} - \mathbf{\Lambda}(\mathbf{u}))\, \text{g}^{-1}(\mathbf{x}). \tag{20}$$

By the change-of-variables formula,

$$\log p_{\boldsymbol{\theta}}(\mathbf{x} \mid \mathbf{u}) = \log p_{\boldsymbol{\theta}}(\mathbf{n} \mid \mathbf{u}) + \log |\det(\mathbf{I} - \boldsymbol{\Lambda}(\mathbf{u}))| + \log \left| \det J_{\mathrm{g}^{-1}}(\mathbf{x}) \right|$$
$$= \log p_{\boldsymbol{\theta}}(\mathbf{n} \mid \mathbf{u}) + \log \left| \det J_{\mathrm{g}^{-1}}(\mathbf{x}) \right|, \tag{21}$$

where the last equality uses $\left| \det(\mathbf{I} - \boldsymbol{\Lambda}(\mathbf{u})) \right| = 1$.

We assume that, in the limit of infinite data, any two parameterizations $\boldsymbol{\theta}$ and $\hat{\boldsymbol{\theta}}$ from the same model class induce the same conditional distribution over the observations, i.e.,

$$p_{\boldsymbol{\theta}}(\mathbf{x} \mid \mathbf{u}) \equiv p_{\hat{\boldsymbol{\theta}}}(\mathbf{x} \mid \mathbf{u}) \qquad \text{for all } (\mathbf{x}, \mathbf{u}). \tag{22}$$

Applying Eq. 21 to $\hat{\boldsymbol{\theta}}$ yields

$$\log p_{\hat{\boldsymbol{\theta}}}(\mathbf{x} \mid \mathbf{u}) = \log p_{\hat{\boldsymbol{\theta}}}(\hat{\mathbf{n}} \mid \mathbf{u}) + \log \left| \det J_{\hat{\mathrm{g}}^{-1}}(\mathbf{x}) \right|. \tag{23}$$

Subtracting these two expressions and using Eq. 22 gives

$$\log p_{\boldsymbol{\theta}}(\mathbf{n} \mid \mathbf{u}) - \log p_{\hat{\boldsymbol{\theta}}}(\hat{\mathbf{n}} \mid \mathbf{u}) = \log \left| \det J_{\hat{\mathrm{g}}^{-1}}(\mathbf{x}) \right| - \log \left| \det J_{\mathrm{g}^{-1}}(\mathbf{x}) \right| =: \boldsymbol{\phi}(\mathbf{x}), \tag{24}$$

where $\boldsymbol{\phi}(\mathbf{x})$ does not depend on $\mathbf{u}$.

For any environment $\mathbf{u}_\ell$, subtracting Eq. 24 at $\mathbf{u}_0$ from that at $\mathbf{u}_\ell$ gives

$$\log \frac{p_{\boldsymbol{\theta}}(\mathbf{n} \mid \mathbf{u}_\ell)}{p_{\boldsymbol{\theta}}(\mathbf{n} \mid \mathbf{u}_0)} = \log \frac{p_{\hat{\boldsymbol{\theta}}}(\hat{\mathbf{n}} \mid \mathbf{u}_\ell)}{p_{\hat{\boldsymbol{\theta}}}(\hat{\mathbf{n}} \mid \mathbf{u}_0)}.$$

Since $p_{\boldsymbol{\theta}}(\mathbf{n} \mid \mathbf{u}) = p(\mathbf{n}_\iota)\, p(\mathbf{n}_\nu \mid \mathbf{u})$ and $p(\mathbf{n}_\iota)$ is invariant in $\mathbf{u}$, the $\mathbf{n}_\iota$-terms cancel from the left side, yielding

$$\log \frac{p(\mathbf{n}_\nu \mid \mathbf{u}_\ell)}{p(\mathbf{n}_\nu \mid \mathbf{u}_0)} = \log \frac{p_{\hat{\boldsymbol{\theta}}}(\hat{\mathbf{n}} \mid \mathbf{u}_\ell)}{p_{\hat{\boldsymbol{\theta}}}(\hat{\mathbf{n}} \mid \mathbf{u}_0)}. \tag{25}$$

**Step 2:** Starting from the ratio identity Eq. 25 and using the diagonal Gaussian log-density

$$\log p(\mathbf{n}_\nu \mid \mathbf{u}) = -\frac{1}{2} \sum_{j=1}^{d_\nu} \left[ \log \beta_{\nu,j}(\mathbf{u}) + \frac{(n_{\nu,j} - \mu_{\nu,j}(\mathbf{u}))^2}{\beta_{\nu,j}(\mathbf{u})} \right] + C, \tag{26}$$

a direct expansion of the quadratic term shows that Eq. 25 is equivalent, for each $\ell = 1, \ldots, 2d_\nu$, to

$$\Delta\boldsymbol{\eta}(\mathbf{u}_\ell)^\top \underbrace{\begin{pmatrix} \mathbf{n}_\nu \\ \mathbf{n}_\nu^{\odot 2} \end{pmatrix}}_{=:\mathbf{T}_\nu(\mathbf{n}_\nu)} = \Delta\hat{\boldsymbol{\eta}}(\mathbf{u}_\ell)^\top \underbrace{\begin{pmatrix} \hat{\mathbf{n}} \\ \hat{\mathbf{n}}^{\odot 2} \end{pmatrix}}_{=:\hat{\mathbf{T}}(\hat{\mathbf{n}})} + b_\ell, \tag{27}$$

where $\Delta\eta(\cdot)$ and $\Delta\hat{\eta}(\cdot)$ collect the environment-dependent coefficients, and $b_\ell$ is a scalar independent of $\mathbf{n}_\nu$.

Stacking Eq. 27 over $\ell = 1, \ldots, 2d_\nu$ yields

$$\mathbf{L}^\top T_\nu(\mathbf{n}_\nu) = \hat{\mathbf{L}}^\top \hat{T}(\hat{\mathbf{n}}) + \mathbf{b}. \tag{28}$$

By the full-rank assumption on $\mathbf{L}^\top$, left-multiplying Eq. 28 by a left inverse of $\mathbf{L}^\top$ yields

$$\mathbf{T}_\nu(\mathbf{n}_\nu) = \mathbf{A}_\nu \hat{\mathbf{T}}(\hat{\mathbf{n}}) + \mathbf{c}_\nu. \tag{29}$$

We now argue that $\mathbf{A}_\nu \in \mathbb{R}^{2d_\nu \times 2(d_\iota + d_\nu)}$ has full row rank. Since $\mathbf{T}_\nu(\mathbf{n}_\nu) = (\mathbf{n}_\nu, \mathbf{n}_\nu^{\odot 2})$ is component-wise and consists of $k = 2$ univariate sufficient statistics per coordinate, the standard rank argument for nonlinear ICA with exponential family conditionals applies. In particular, by the construction in Appendix A of Sorrenson et al. (2020), there exist $k = 2$ evaluation points at which the concatenated Jacobians of $T_\nu$ form an invertible matrix, which implies that the linear operator relating sufficient statistics must be full rank. We therefore conclude that $\mathbf{A}_\nu$ has full row rank.

**Step 3:** Eq. 29 implies that each $n_{\nu,j}$ is a polynomial of degree at most 2 in $\hat{\mathbf{n}}$, and simultaneously $n_{\nu,j}^2$ is also a polynomial of degree at most 2 in $\hat{\mathbf{n}}$. Since the coordinates of $\mathbf{n}_\nu$ are mutually independent and Gaussian, any cross-dependence on multiple coordinates of $\hat{\mathbf{n}}$ would introduce statistical dependence, yielding a contradiction. Therefore each $n_{\nu,j}$ depends on at most one coordinate of $\hat{\mathbf{n}}$, and hence there exist a permutation $\pi$, nonzero scalars $a_j$, and constants $b_j$ such that $n_{\nu,j} = a_j \hat{n}_{\pi(j)} + b_j$ for all $j$; see, e.g., Sorrenson et al. (2020). $\qquad \square$

**Lemma D.3** (Block Identifiability of $\mathbf{n}_\iota$ via Alignment)**.** *Write* $\mathbf{f}(\mathbf{x}) = (\mathbf{f}_\iota(\mathbf{x}), \mathbf{f}_\nu(\mathbf{x}))$, *where* $\mathbf{f}_\iota$ *is the* $d_\iota$-*dimensional component intended to estimate the invariant noise* $\mathbf{n}_\iota$, *denoted by* $\hat{\mathbf{n}}_\iota$. *Consider the latent generative model in Eqs. 6–9. Let g be the true decoder and* $f = \hat{g}^{-1}$ *be the estimated encoder, both smooth and invertible.*

*Assume that for any fixed* $\mathbf{n}_\iota$ *and any environments* $\mathbf{u}, \mathbf{u}_0$, *if* $\mathbf{N}_\nu^{(\mathbf{u})}$ *and* $\mathbf{N}_\nu^{(\mathbf{u}_0)}$ *are independent draws from the corresponding noise distributions, then*

$$\mathbf{f}_\iota\big(g(\mathbf{z}(\mathbf{n}_\iota, \mathbf{N}_\nu^{(\mathbf{u})}, \mathbf{u})))\big) = \mathbf{f}_\iota\big(g(\mathbf{z}(\mathbf{n}_\iota, \mathbf{N}_\nu^{(\mathbf{u}_0)}, \mathbf{u}_0)))\big) \quad a.s. \tag{30}$$

*Then there exists a smooth function* $\mathbf{h}_\iota$ *such that*

$$\mathbf{f}_\iota\big(g(\mathbf{z}(\mathbf{n}_\iota, \mathbf{n}_\nu, \mathbf{u})))\big) = \mathbf{h}_\iota(\mathbf{n}_\iota) \quad a.s., \tag{31}$$

*i.e.,* $\hat{\mathbf{n}}_\iota$ *does not depend on* $\mathbf{n}_\nu$.

*Proof.* Define the deterministic map

$$F(\mathbf{n}_\iota, \mathbf{n}_\nu, \mathbf{u}) := \mathbf{f}_\iota\big(g(\mathbf{z}(\mathbf{n}_\iota, \mathbf{n}_\nu, \mathbf{u}))\big), \tag{32}$$

where $\mathbf{z}(\cdot)$ denotes the unique solution of the linear structural equations given $(\mathbf{n}_\iota, \mathbf{n}_\nu, \mathbf{u})$. Since $\mathbf{f}$, g are smooth and invertible and $\mathbf{z}(\cdot)$ is linear in $(\mathbf{n}_\iota, \mathbf{n}_\nu)$, the map $F$ is smooth in $(\mathbf{n}_\iota, \mathbf{n}_\nu)$ for each fixed $\mathbf{u}$.

By assumption Eq. 30, for any fixed $\mathbf{n}_\iota$ and any $\mathbf{u}, \mathbf{u}_0$, the random variables $F(\mathbf{n}_\iota, \mathbf{N}_\nu^{(\mathbf{u})}, \mathbf{u})$ and $F(\mathbf{n}_\iota, \mathbf{N}_\nu^{(\mathbf{u}_0)}, \mathbf{u}_0)$ are almost surely equal. In particular, for each fixed $\mathbf{n}_\iota$ and $\mathbf{u}$, the random variable $F(\mathbf{n}_\iota, \mathbf{N}_\nu^{(\mathbf{u})}, \mathbf{u})$ is almost surely constant with respect to $\mathbf{N}_\nu^{(\mathbf{u})}$.

We show that $F(\mathbf{n}_\iota, \mathbf{n}_\nu, \mathbf{u})$ cannot depend on $\mathbf{n}_\nu$. Suppose otherwise. Then there exists an environment $\mathbf{u}$ and an index $l \in \{1, \dots, d_\nu\}$ such that

$$\frac{\partial F}{\partial n_{\nu,l}}(\mathbf{n}_\iota^\star, \mathbf{n}_\nu^\star, \mathbf{u}) \neq 0 \tag{33}$$

for some $(\mathbf{n}_\iota^\star, \mathbf{n}_\nu^\star)$. By continuity of the partial derivative, there exists an open neighborhood $U = U_\iota \times U_\nu$ of $(\mathbf{n}_\iota^\star, \mathbf{n}_\nu^\star)$ on which $\partial F / \partial n_{\nu,l}$ has a fixed nonzero sign.

Fix any $\mathbf{n}_\iota \in U_\iota$ and $\mathbf{n}_{\nu,-l} \in U_{\nu,-l}$. Then the function

$$t \mapsto F(\mathbf{n}_\iota, (t, \mathbf{n}_{\nu,-l}), \mathbf{u}) \tag{34}$$

is strictly monotone on $U_{\nu,l}$. Since $\mathbf{N}_\nu^{(\mathbf{u})}$ has a non-degenerate Gaussian distribution, $\mathbb{P}(\mathbf{N}_\nu^{(\mathbf{u})} \in U_\nu) > 0$, and conditional on this event, two independent draws have distinct $l$-th coordinates with probability one. Therefore,

$$\mathbb{P}\Big(F(\mathbf{n}_\iota, \mathbf{N}_\nu^{(\mathbf{u})}, \mathbf{u}) \neq F(\mathbf{n}_\iota, \tilde{\mathbf{N}}_\nu^{(\mathbf{u})}, \mathbf{u})\Big) > 0, \tag{35}$$

which contradicts the almost-sure invariance implied by Eq. 30. Hence $F$ cannot depend on $\mathbf{n}_\nu$.

Therefore, for each $\mathbf{u}$ there exists a function $\mathbf{h}_{\iota,\mathbf{u}}$ such that $F(\mathbf{n}_\iota, \mathbf{n}_\nu, \mathbf{u}) = \mathbf{h}_{\iota,\mathbf{u}}(\mathbf{n}_\iota)$ almost surely. The alignment equality across environments forces these functions to agree almost surely, yielding a single smooth function $\mathbf{h}_\iota$ such that

$$F(\mathbf{n}_\iota, \mathbf{n}_\nu, \mathbf{u}) = \mathbf{h}_\iota(\mathbf{n}_\iota) \quad \text{a.s. for all } \mathbf{u}. \tag{36}$$

$\square$

**Lemma D.4** (Linear Block Identifiability of $\mathbf{n}_\iota$)**.** *Under the conditions of Lemma D.3 and assuming the exogenous noises* $\mathbf{n}_\iota$ *and* $\hat{\mathbf{n}}_\iota$ *follow non-degenerate Gaussian distributions, the block-wise mapping* $\mathbf{h}_\iota$ *must be an **affine transformation**:*

$$\hat{\mathbf{n}}_\iota = \mathbf{A}_\iota \mathbf{n}_\iota + \mathbf{b}_\iota, \tag{37}$$

*where* $\mathbf{A}_\iota \in \mathbb{R}^{d_\iota \times d_\iota}$ *is a constant non-singular matrix and* $\mathbf{b}_\iota \in \mathbb{R}^{d_\iota}$ *is a bias vector.*

*Proof.* By Lemma D.3, there exists a smooth bijection $\mathbf{h}_\iota : \mathbb{R}^{d_\iota} \to \mathbb{R}^{d_\iota}$ such that $\hat{\mathbf{n}}_\iota = \mathbf{h}_\iota(\mathbf{n}_\iota)$ almost surely. Assume $\mathbf{n}_\iota \sim \mathcal{N}(\boldsymbol{\mu}_\iota, \boldsymbol{\Sigma}_\iota)$ and $\hat{\mathbf{n}}_\iota \sim \mathcal{N}(\hat{\boldsymbol{\mu}}_\iota, \hat{\boldsymbol{\Sigma}}_\iota)$ with $\boldsymbol{\Sigma}_\iota, \hat{\boldsymbol{\Sigma}}_\iota \succ 0$.

Let $p$ and $\hat{p}$ denote the densities of $\mathbf{n}_\iota$ and $\hat{\mathbf{n}}_\iota$, respectively. By change of variables,

$$\log p(\mathbf{n}) = \log \hat{p}(\mathbf{h}_\iota(\mathbf{n})) + \log \big| \det J_{\mathbf{h}_\iota}(\mathbf{n}) \big|. \tag{38}$$

Taking the gradient w.r.t. $\mathbf{n}$ on both sides yields

$$\nabla_{\mathbf{n}} \log p(\mathbf{n}) = J_{\mathbf{h}_\iota}(\mathbf{n})^\top \nabla_{\hat{\mathbf{n}}} \log \hat{p}(\hat{\mathbf{n}})\big|_{\hat{\mathbf{n}} = \mathbf{h}_\iota(\mathbf{n})} + \nabla_{\mathbf{n}} \log \big| \det J_{\mathbf{h}_\iota}(\mathbf{n}) \big|. \tag{39}$$

For a non-degenerate Gaussian, the score function is affine:

$$\nabla_{\mathbf{n}} \log p(\mathbf{n}) = -\boldsymbol{\Sigma}_\iota^{-1}(\mathbf{n} - \boldsymbol{\mu}_\iota), \qquad \nabla_{\hat{\mathbf{n}}} \log \hat{p}(\hat{\mathbf{n}}) = -\hat{\boldsymbol{\Sigma}}_\iota^{-1}(\hat{\mathbf{n}} - \hat{\boldsymbol{\mu}}_\iota).$$

Substituting these into equation 39 gives, for all $\mathbf{n}$,

$$-\boldsymbol{\Sigma}_\iota^{-1}(\mathbf{n} - \boldsymbol{\mu}_\iota) = -J_{\mathbf{h}_\iota}(\mathbf{n})^\top \hat{\boldsymbol{\Sigma}}_\iota^{-1}(\mathbf{h}_\iota(\mathbf{n}) - \hat{\boldsymbol{\mu}}_\iota) + \nabla_{\mathbf{n}} \log \big| \det J_{\mathbf{h}_\iota}(\mathbf{n}) \big|. \tag{40}$$

Now differentiate equation 40 once more w.r.t. $\mathbf{n}$. The left-hand side has constant Jacobian $-\boldsymbol{\Sigma}_\iota^{-1}$. On the right-hand side, any nonzero second derivatives of $\mathbf{h}_\iota$ would generate terms depending on $\mathbf{n}$ through $(\mathbf{h}_\iota(\mathbf{n}) - \hat{\boldsymbol{\mu}}_\iota)$ and through the derivatives of $\log \big| \det J_{\mathbf{h}_\iota}(\mathbf{n}) \big|$. Since the equality holds for all $\mathbf{n}$ and the left-hand side is constant, it follows that $\nabla^2 \mathbf{h}_\iota(\mathbf{n}) \equiv 0$, i.e., $\mathbf{h}_\iota$ has constant Jacobian. Therefore $\mathbf{h}_\iota$ is affine:

$$\mathbf{h}_\iota(\mathbf{n}) = \mathbf{A}_\iota \mathbf{n} + \mathbf{b}_\iota,$$

with $\mathbf{A}_\iota$ nonsingular because $\mathbf{h}_\iota$ is bijective. $\qquad\qquad\square$

# E. Justification and Clarification for Assumptions in Theorem 1

Assumptions (i)-(ii) are originally developed by nonlinear ICA (Hyvarinen & Morioka, 2016; 2017; Khemakhem et al., 2020; Sorrenson et al., 2020), and have also been adopted in several recent works on causal representation learning, with different forms (Zhang et al., 2024; Liu et al., 2026c; 2024b). Intuitively, assumption (i) requires that the mapping from the latent space to the observation space be information-preserving, in the sense that no latent information is irreversibly lost during the generation process. If this condition were violated, exact recovery of the latent variables would in general be impossible, regardless of the learning algorithm. Assumption (ii) ensures that the auxiliary variable (environment or intervention) induces sufficiently significant and diverse changes in the distributions of the latent components. This condition guarantees that different latent factors respond distinctly across environments, providing the variation for disentanglement.

Assumption (iii) is inspired by analyses in contrastive and invariant representation learning, where optimal alignment objectives are assumed to recover representations that are invariant across different views or environments. Related theoretical analyses have been provided in recent contrastive learning works (Wang & Isola, 2020; Saunshi et al., 2019; Von Kügelgen et al., 2021). In particular, Von Kügelgen et al. (2021) (e..g, Theorems 4.3 and 4.4) explicitly adopt a global optimality assumption to establish identifiability guarantees for self-supervised representations. Following this line of work, we likewise impose a population-level optimality assumption to connect the alignment objective with identifiability. We emphasize that such optimality assumptions are standard in the identifiability literature, and are commonly introduced to characterize what is theoretically achievable under idealized conditions, rather than to describe practical optimization behavior (Hyvarinen & Morioka, 2016; 2017; Khemakhem et al., 2020; Sorrenson et al., 2020; Von Kügelgen et al., 2021).

Assumption (iv) is aligned with identifiability analyses in causal representation learning, where variations across environments are required to disentangle causal mechanisms (Lippe et al., 2022b;a; Brehmer et al., 2022; Liu et al., 2026c; 2024b; von Kügelgen et al., 2023). Intuitively, this assumption requires the existence of at least one environment $\mathbf{u}$ in which the influence of parent variables on a given node is effectively removed, thereby isolating the corresponding latent noise component. This can be interpreted as a hard intervention on the target variable, and is introduced to rule out degenerate causal structures in which parent effects cannot be separated from intrinsic noise.

# F. Proof of Theorem 1

Based on the preceding lemmas, we now establish the following identifiability results for the latent variables $\mathbf{z} = (\mathbf{z}_\iota, \mathbf{z}_\nu)$.

**Theorem 1** (Identifiability Results). *Suppose the observed variable* $\mathbf{x}$ *and latent causal variables* $\mathbf{z} = (\mathbf{z}_\iota, \mathbf{z}_\nu)$ *follow the generative model defined in Eqs. 6–9, parameterized by* $\boldsymbol{\theta} = (g, \boldsymbol{\lambda}, \boldsymbol{\mu}, \boldsymbol{\beta})$. *Let* $\hat{\boldsymbol{\theta}} = (\hat{g}, \hat{\boldsymbol{\lambda}}, \hat{\boldsymbol{\mu}}, \hat{\boldsymbol{\beta}})$ *be the estimated parameters obtained by matching the conditional data distribution* $p_\theta(\mathbf{x}|\mathbf{u}) = p_{\hat{\theta}}(\mathbf{x}|\mathbf{u})$ *and minimizing the alignment loss. Assume the following conditions hold:*

(i) ***Invertibility and Smoothness:*** *The unknown nonlinear mapping* $g$ *is smooth and invertible.*

(ii) ***Environmental Sufficiency:*** *There exist* $2d_\nu$ *distinct environments* $\{\mathbf{u}_1, \ldots, \mathbf{u}_m\}$ *relative to a reference* $\mathbf{u}_0$ *such that the matrix*

$$\mathbf{L}^\top = [\Delta\boldsymbol{\eta}(\mathbf{u}_1), \ldots, \Delta\boldsymbol{\eta}(\mathbf{u}_{2d_\nu})]^\top \in \mathbb{R}^{2d_\nu \times 2d_\nu} \tag{41}$$

*has full column rank* $2d_\nu$, *where (elementwise divisions)*

$$\Delta\eta(\mathbf{u}) := \begin{pmatrix} \frac{\boldsymbol{\mu}_\nu(\mathbf{u})}{\boldsymbol{\beta}_\nu(\mathbf{u})} - \frac{\boldsymbol{\mu}_\nu(\mathbf{u}_0)}{\boldsymbol{\beta}_\nu(\mathbf{u}_0)} \\ -\frac{1}{2}\left(\frac{1}{\boldsymbol{\beta}_\nu(\mathbf{u})} - \frac{1}{\boldsymbol{\beta}_\nu(\mathbf{u}_0)}\right) \end{pmatrix} \in \mathbb{R}^{2d_\nu}. \tag{42}$$

.

(iii) ***Optimal Alignment:*** *The alignment loss, e.g., Eq. 4 attains its global minimum such that* $\mathbf{f}_\iota(\mathbf{x}^{(\mathbf{u})}) = \mathbf{f}_\iota(\mathbf{x}^{(\mathbf{u}_0)})$ *almost surely for any* $\mathbf{u}, \mathbf{u}_0$, *where* $\mathbf{f}_\iota = \hat{g}_\iota^{-1}$.

(iv) ***Intervention Sufficiency:*** *The function class of* $\boldsymbol{\lambda}$ *satisfies the following condition: there exists* $\mathbf{u}_i$, *such that, for all parent nodes* $z_j \in \text{pa}_i$ *of* $z_i$, $\boldsymbol{\lambda}_{j,i} = 0$.

*Then, the true latent causal variables* $\mathbf{z}$ *are related to the variables* $\hat{\mathbf{z}}$ *estimated by matching likelihood (i.e., the upper bound of ELBO Eq. 3) as follows:*

1. $\mathbf{z}_\nu$ *is identified up to permutation and scaling, i.e.,* $\mathbf{z}_\nu = \mathbf{P}_\nu\hat{\mathbf{z}}_\nu + \mathbf{c}_\nu$, *where* $\mathbf{P}_\nu$ *is a permutation matrix with scaling.*

2. $\mathbf{z}_\iota$ *is identified up to a linear block transformation, i.e.,* $\mathbf{z}_\iota = \mathbf{A}_\iota\hat{\mathbf{z}}_\iota + \mathbf{c}_\iota$, *where* $\mathbf{A}_\iota \in \mathbb{R}^{d_\iota \times d_\iota}$ *is a non-singular matrix.*

*Proof.* **Step 1: Identification Results of Latent Noise n.** From Lemma D.2, the environmental noise $\mathbf{n}_\nu$ is identified component-wise due to sufficient variance across environments: $\mathbf{n}_\nu = \mathbf{P}_n\hat{\mathbf{n}}_\nu + \mathbf{b}_\nu$, where $\mathbf{P}_n$ is a permutation matrix with scaling. From Lemma D.4, the invariant noise $\mathbf{n}_\iota$ is identified as a linear block: $\mathbf{n}_\iota = \mathbf{A}_n\hat{\mathbf{n}}_\iota + \mathbf{b}_\iota$. Combining these, the global exogenous noise vector satisfies:

$$\mathbf{n} = \mathbf{A}_{global}\hat{\mathbf{n}} + \mathbf{b}, \quad \text{with } \mathbf{A}_{global} = \begin{pmatrix} \mathbf{A}_n & \mathbf{0} \\ \mathbf{0} & \mathbf{P}_n \end{pmatrix}. \tag{43}$$

**Step 2: Derivation of Linear Latent Relationship.** Substituting the structural equations $\mathbf{n} = (\mathbf{I} - \boldsymbol{\Lambda}(\mathbf{u}))\mathbf{z}$ and $\hat{\mathbf{n}} = (\mathbf{I} - \hat{\boldsymbol{\Lambda}}(\mathbf{u}))\hat{\mathbf{z}}$ (See Eq. 13) into the noise relationship yields:

$$(\mathbf{I} - \boldsymbol{\Lambda}(\mathbf{u}))\mathbf{z} = \mathbf{A}_{global}(\mathbf{I} - \hat{\boldsymbol{\Lambda}}(\mathbf{u}))\hat{\mathbf{z}} + \mathbf{b}_n. \tag{44}$$

Solving Eq. 44 for $\mathbf{z}$ thus implies a constant affine mapping:

$$\mathbf{z} = \mathbf{M}\hat{\mathbf{z}} + \mathbf{c}, \tag{45}$$

where the transformation matrix

$$\mathbf{M} = (\mathbf{I} - \boldsymbol{\Lambda}(\mathbf{u}))^{-1}\mathbf{A}_{global}(\mathbf{I} - \hat{\boldsymbol{\Lambda}}(\mathbf{u})). \tag{46}$$

Since $\mathbf{z} = g^{-1}(\mathbf{x})$ and $\hat{\mathbf{z}} = \hat{g}^{-1}(\mathbf{x})$, the mapping between $\mathbf{z}$ and $\hat{\mathbf{z}}$ is given by $\mathbf{z} = (g^{-1} \circ \hat{g})(\hat{\mathbf{z}})$. Because both $g$ and $\hat{g}$ are assumed to be independent of the environment $\mathbf{u}$, the transformation from $\hat{\mathbf{z}}$ to $\mathbf{z}$ must also be $\mathbf{u}$-invariant. As a result, $\mathbf{M}$ is independent of $\mathbf{u}$.

**Step 3: Identifiability Results of Latent z.**

**Linear Block Identifiability of $\mathbf{z}_\iota$.** From Lemma D.4, there exist a non-singular matrix $\mathbf{A}_\iota$ and a vector $\mathbf{b}_\iota$ such that

$$\hat{\mathbf{n}}_\iota = \mathbf{A}_\iota \mathbf{n}_\iota + \mathbf{b}_\iota. \tag{47}$$

By the structural equations for the invariant block,

$$\mathbf{n}_\iota = (\mathbf{I} - \boldsymbol{\lambda}_{\iota\iota})\mathbf{z}_\iota, \qquad \hat{\mathbf{n}}_\iota = (\mathbf{I} - \hat{\boldsymbol{\lambda}}_{\iota\iota})\hat{\mathbf{z}}_\iota. \tag{48}$$

Combining these yields

$$(\mathbf{I} - \hat{\boldsymbol{\lambda}}_{\iota\iota})\hat{\mathbf{z}}_\iota = \mathbf{A}_\iota(\mathbf{I} - \boldsymbol{\lambda}_{\iota\iota})\mathbf{z}_\iota + \mathbf{b}_\iota, \tag{49}$$

which implies that $\mathbf{z}_\iota$ is identifiable up to an invertible linear transformation (and a shift).

**Component-Wise Identifiability of $\mathbf{z}_\nu$** To determine the internal structure of $\mathbf{z}_\nu$, we temporarily treat the invariant block $\mathbf{z}_\iota$ as an aggregated variable and do not resolve its internal coordinates. That is, for the purpose of analyzing the $\nu$-block, we regard $\mathbf{z}_\iota$ as a single latent variable $z_\iota$ and correspondingly $\mathbf{n}_\iota$ as a single noise variable $n_\iota$.

Under this reduced representation, the block-wise linear relation $\hat{\mathbf{n}}_\iota = \mathbf{A}_\iota \mathbf{n}_\iota + \mathbf{b}_\iota$ from Lemma D.4 reduces to a scalar affine relation $\hat{n}_\iota = s\, n_\iota + b$ for some $s \neq 0$.

Consequently, in this reduced view, the global linear transformation in Eq. 43 takes the form

$$\mathbf{A}'_{\text{global}} = \begin{pmatrix} s & \mathbf{0} \\ \mathbf{0} & \mathbf{P}_n \end{pmatrix}, \tag{50}$$

where $\mathbf{P}_n$ is a diagonal scaling-permutation matrix on the $\nu$-coordinates.

Therefore, by Lemma D.2, the reduced noise vector $(n_\iota, \mathbf{n}_\nu)$ is identified up to a joint permutation and component-wise scaling, and in particular $\mathbf{n}_\nu$ is identified up to permutation and scaling independently of the internal parameterization of $\mathbf{n}_\iota$.

Then, under Assumption (iv), the proof can be completed by following Step III in the proof of Theorem 1 of Liu et al. (2026c). For completeness, we briefly adapt the argument to our setting below. The key idea in Step III is to examine the block structure of the linear mapping between the true and estimated latents, exploiting the fact that (i) $\mathbf{I} - \boldsymbol{\Lambda}(\mathbf{u})$ and $\mathbf{I} - \hat{\boldsymbol{\Lambda}}(\mathbf{u})$ are strictly lower triangular with unit diagonal, and (ii) the $\nu$-block of the noise transformation is diagonal up to permutation and scaling by Lemma D.2. By comparing corresponding entries of the resulting matrix identity across carefully chosen environments where individual structural coefficients vanish (as guaranteed by Assumption (iv)), one shows that all off-diagonal entries of the $\nu\nu$ submatrix must be zero, yielding identification of $\mathbf{z}_\nu$ up to permutation and scaling. $\qquad\square$

# G. Implementation of the Evidence Lower Bound

In this appendix, we provide a general derivation of the Evidence Lower Bound (ELBO) for our generative model, valid for any intervention vector $\mathbf{u}$.

**Generative Model.** For an observation $\mathbf{x}$ under intervention $\mathbf{u}$, the generative model factorizes as:

$$p(\mathbf{x}, \mathbf{z}_\nu, \mathbf{z}_\iota \mid \mathbf{u}) = p(\mathbf{x} \mid \mathbf{z}_\nu, \mathbf{z}_\iota)\, p(\mathbf{z}_\nu \mid \mathbf{u}, \mathbf{z}_\iota)\, p(\mathbf{z}_\iota), \tag{51}$$

where $\mathbf{z}_\nu$ denotes the *variant* (intervention-specific) latents and $\mathbf{z}_\iota$ the *invariant* latents. The variational posterior adopts the structured mean-field factorization:

$$q(\mathbf{z}_\nu, \mathbf{z}_\iota \mid \mathbf{x}, \mathbf{u}) = q(\mathbf{z}_\nu \mid \mathbf{x}, \mathbf{u})\, q(\mathbf{z}_\iota \mid \mathbf{x}). \tag{52}$$

**Derivation.** The marginal likelihood is

$$\log p(\mathbf{x} \mid \mathbf{u}) = \log \int \frac{p(\mathbf{x}, \mathbf{z}_\nu, \mathbf{z}_\iota \mid \mathbf{u})}{q(\mathbf{z}_\nu, \mathbf{z}_\iota \mid \mathbf{x}, \mathbf{u})} q(\mathbf{z}_\nu, \mathbf{z}_\iota \mid \mathbf{x}, \mathbf{u})\, d\mathbf{z}_\nu d\mathbf{z}_\iota.$$

Applying Jensen's inequality to the logarithm yields the ELBO:

$$\mathcal{L}_{\text{ELBO}}(\mathbf{x}, \mathbf{u}) = \mathbb{E}_{q(\mathbf{z}_\nu, \mathbf{z}_\iota \mid \mathbf{x}, \mathbf{u})}\big[\log p(\mathbf{x} \mid \mathbf{z}_\nu, \mathbf{z}_\iota)\big] - D_{\text{KL}}(q(\mathbf{z}_\nu \mid \mathbf{x}, \mathbf{u}) \,\|\, p(\mathbf{z}_\nu \mid \mathbf{u}, \mathbf{z}_\iota)) - D_{\text{KL}}(q(\mathbf{z}_\iota \mid \mathbf{x}) \,\|\, p(\mathbf{z}_\iota)). \tag{53}$$

Considering the trade-off between reconstruction and regularization and motivated by $\beta$-VAE, we implement the ELBO as

$$\mathcal{L}_{\text{ELBO}}(\mathbf{x}, \mathbf{u}) = \mathbb{E}_{q(\mathbf{z}_\nu, \mathbf{z}_\iota \mid \mathbf{x}, \mathbf{u})}\big[\log p_\phi(\mathbf{x} \mid \mathbf{z}_\nu, \mathbf{z}_\iota)\big] - \beta_\nu\, D_{\text{KL}}(q_\theta(\mathbf{z}_\nu \mid \mathbf{x}, \mathbf{u}) \,\|\, p_\phi(\mathbf{z}_\nu \mid \mathbf{u}, \mathbf{z}_\iota)) - \beta_\iota\, D_{\text{KL}}(q_\theta(\mathbf{z}_\iota \mid \mathbf{x}) \,\|\, p_\phi(\mathbf{z}_\iota)). \tag{54}$$

**Priors.** Following the latent causal generative model Eqs. 6-8 in theoretical analysis in Sec. 4, we implement the priors appearing in the ELBO in Eq. 53 as follows: we specify (i) a Gaussian prior for the invariant block $\mathbf{z}_\iota$, and (ii) a DAG-structured linear-Gaussian conditional prior for the perturbation-responsive block $\mathbf{z}_\nu$ conditioned on the intervention label $\mathbf{u}$ and $\mathbf{z}_\iota$.

**Invariant Prior $p(\mathbf{z}_\iota)$.** We place a standard diagonal Gaussian prior on the invariant latent variables:

$$p(\mathbf{z}_\iota) = \mathcal{N}(\mathbf{0}, \mathbf{I}). \tag{55}$$

This choice matches the third term in Eq. 53 and yields a closed-form KL divergence with the diagonal-Gaussian variational posterior $q(\mathbf{z}_\iota \mid \mathbf{x})$.

**Variant Prior $p(\mathbf{z}_\nu \mid \mathbf{u}, \mathbf{z}_\iota)$.** Following the linear structural causal model in Eqs. 6–8, we parameterize the conditional prior over $\mathbf{z}_\nu$ via a DAG-structured linear-Gaussian model with a fixed causal ordering. Let $\boldsymbol{\Lambda}_{\nu\nu}(\mathbf{u})$ be strictly lower triangular (acyclic), and define

$$\mathbf{A}(\mathbf{u}) := \mathbf{I} - \boldsymbol{\Lambda}_{\nu\nu}(\mathbf{u}). \tag{56}$$

We implement latent noise

$$\mathbf{n}_\nu \sim \mathcal{N}\big(\boldsymbol{\mu}_\nu(\mathbf{u}), \text{diag}(\boldsymbol{\beta}_\nu(\mathbf{u}))\big), \tag{57}$$

and a linear dependence on $\mathbf{z}_\iota$ through $\boldsymbol{\Lambda}_{\nu\iota}(\mathbf{u})$:

$$\mathbf{z}_\nu = \boldsymbol{\Lambda}_{\nu\nu}(\mathbf{u})\, \mathbf{z}_\nu + \boldsymbol{\Lambda}_{\nu\iota}(\mathbf{u})\, \mathbf{z}_\iota + \mathbf{n}_\nu. \tag{58}$$

Equivalently, since $\mathbf{A}(\mathbf{u})$ is invertible under acyclicity, this induces the conditional Gaussian prior

$$p(\mathbf{z}_\nu \mid \mathbf{u}, \mathbf{z}_\iota) = \mathcal{N}\Big(\mathbf{A}(\mathbf{u})^{-1}\big(\boldsymbol{\Lambda}_{\nu\iota}(\mathbf{u})\, \mathbf{z}_\iota + \boldsymbol{\mu}_\nu(\mathbf{u})\big),\ \mathbf{A}(\mathbf{u})^{-1} \text{diag}(\boldsymbol{\beta}_\nu(\mathbf{u}))\, \mathbf{A}(\mathbf{u})^{-\top}\Big). \tag{59}$$

In practice, we implement this prior by sampling $\mathbf{n}_\nu$ and solving the triangular linear system in Eq. 58, which avoids explicit matrix inversion and naturally enforces the DAG constraint via the strictly lower-triangular $\boldsymbol{\Lambda}_{\nu\nu}(\mathbf{u})$.

---

**Algorithm 1** Training Procedure of PerturbedVAE

---

**Require:** Dataset $\mathcal{D}$
1: $(\mathbf{x}, \mathbf{u}, \mathbf{x}^{(\mathbf{u}_0)}) \sim \mathcal{D}$
2: $\mathbf{h}_1 \leftarrow f_{\text{enc}}(\mathbf{x}); \quad \mathbf{h}_2 \leftarrow f_{\text{enc}}(\mathbf{x}^{(\mathbf{u}_0)})$
3: — *Step 1: Encode variational posteriors* —
4: $(\boldsymbol{\mu}'_\nu, \log \boldsymbol{\sigma}'_\nu{}^2) \leftarrow g_\nu(\mathbf{h}_1, \mathbf{u})$
5: $(\boldsymbol{\mu}'_{\iota,1}, \log \boldsymbol{\sigma}'_{\iota,1}{}^2) \leftarrow g_\iota(\mathbf{h}_1); \quad (\boldsymbol{\mu}'_{\iota,2}, \log \boldsymbol{\sigma}'_{\iota,2}{}^2) \leftarrow g_\iota(\mathbf{h}_2)$
6: $\boldsymbol{\varepsilon}_\nu, \boldsymbol{\varepsilon}_{\iota,1}, \boldsymbol{\varepsilon}_{\iota,2} \sim \mathcal{N}(\mathbf{0}, \mathbf{I})$
7: $\tilde{\mathbf{z}}_\nu \leftarrow \boldsymbol{\mu}'_\nu + \boldsymbol{\sigma}'_\nu \odot \boldsymbol{\varepsilon}_\nu$                     (sample base noise for $\mathbf{z}_\nu$)
8: $\mathbf{z}_\iota^{(1)} \leftarrow \boldsymbol{\mu}'_{\iota,1} + \boldsymbol{\sigma}'_{\iota,1} \odot \boldsymbol{\varepsilon}_{\iota,1}$
9: $\mathbf{z}_\iota^{(2)} \leftarrow \boldsymbol{\mu}'_{\iota,2} + \boldsymbol{\sigma}'_{\iota,2} \odot \boldsymbol{\varepsilon}_{\iota,2}$
10: — *Step 2: DAG-structured transformation for* $\mathbf{z}_\nu$ —
11: $\boldsymbol{\Lambda}_{\nu\nu}(\mathbf{u}) \leftarrow f_{\nu\nu}(\mathbf{u})$                     (strictly lower-triangular)
12: $\boldsymbol{\Lambda}_{\nu\iota}(\mathbf{u}) \leftarrow f_{\nu\iota}(\mathbf{u})$
13: $\boldsymbol{\mu}_\nu(\mathbf{u}) \leftarrow f_\mu(\mathbf{u})$
14: $\mathbf{z}_\nu \leftarrow \left(\mathbf{I} - \boldsymbol{\Lambda}_{\nu\nu}(\mathbf{u})\right)^{-1}\left(\tilde{\mathbf{z}}_\nu + \boldsymbol{\Lambda}_{\nu\iota}(\mathbf{u})\mathbf{z}_\iota^{(1)} + \boldsymbol{\mu}_\nu(\mathbf{u})\right)$     (implements Eq. equation 58)
15: — *Step 3: Reconstruction* —
16: $\hat{\mathbf{x}} \leftarrow f_{\text{dec}}([\mathbf{z}_\nu, \mathbf{z}_\iota^{(1)}])$
17: — *Step 4: Losses (ELBO + contrastive alignment)* —
18: $\mathcal{L}_{\text{rec}} \leftarrow \|\mathbf{x} - \hat{\mathbf{x}}\|_2^2$
19: $\mathcal{L}_{\text{KL-}\nu} \leftarrow D_{\text{KL}}(q(\mathbf{z}_\nu \mid \mathbf{x}, \mathbf{u}) \,\|\, p(\mathbf{z}_\nu \mid \mathbf{u}, \mathbf{z}_\iota))$
20: $\mathcal{L}_{\text{KL-}\iota} \leftarrow D_{\text{KL}}(q(\mathbf{z}_\iota \mid \mathbf{x}) \,\|\, p(\mathbf{z}_\iota)) + D_{\text{KL}}\left(q(\mathbf{z}_\iota \mid \mathbf{x}^{(\mathbf{u}_0)}) \,\|\, p(\mathbf{z}_\iota)\right)$
21: $\mathcal{L}_{\text{contrast}} \leftarrow \|\boldsymbol{\mu}'_{\iota,1} - \boldsymbol{\mu}'_{\iota,2}\|_2^2$
22: $(\beta_\nu, \beta_\iota, \alpha) \leftarrow \text{Schedule}(t)$                     (annealing schedule)
23: $\mathcal{L}_{\text{total}} \leftarrow \mathcal{L}_{\text{rec}} + \beta_\nu \mathcal{L}_{\text{KL-}\nu} + \beta_\iota \mathcal{L}_{\text{KL-}\iota} + \alpha \mathcal{L}_{\text{contrast}}$
24: Update $\Theta \leftarrow \Theta - \eta \nabla_\Theta \mathcal{L}_{\text{total}}$

---

**Variational Posteriors.** We use a structured variational family that mirrors the decomposition of invariant and perturbation-responsive factors in Eq. 52.

**Invariant Posterior** $q(\mathbf{z}_\iota \mid \mathbf{x})$. The invariant block is parameterized as a diagonal Gaussian:

$$q(\mathbf{z}_\iota \mid \mathbf{x}) = \mathcal{N}\big(\boldsymbol{\mu}'_\iota(\mathbf{x}), \ \text{diag}(\boldsymbol{\sigma}'_\iota(\mathbf{x}))\big), \tag{60}$$

where $(\boldsymbol{\mu}'_\iota(\cdot), \boldsymbol{\sigma}'_\iota(\cdot))$ are outputs of an inference network.

**Variant Posterior** $q(\mathbf{z}_\nu \mid \mathbf{x}, \mathbf{u})$. To capture the causal dependencies among perturbation-responsive variables, we adopt an autoregressive (DAG-ordered) variational posterior:

$$q(\mathbf{z}_\nu \mid \mathbf{x}, \mathbf{u}) = \prod_{i=1}^{d_\nu} q(z_{\nu,i} \mid \mathbf{z}_{\nu,<i}, \mathbf{x}, \mathbf{u}), \tag{61}$$

with each conditional factor being Gaussian,

$$q(z_{\nu,i} \mid \mathbf{z}_{\nu,<i}, \mathbf{x}, \mathbf{u}) = \mathcal{N}\Big(\mu'_{\nu,i}(\mathbf{x}, \mathbf{u}, \mathbf{z}_{\nu,<i}), \ \sigma'_{\nu,i}(\mathbf{x}, \mathbf{u}, \mathbf{z}_{\nu,<i})\Big). \tag{62}$$

This form allows the posterior to represent nontrivial correlations in $\mathbf{z}_\nu$ induced by the directed dependencies, while maintaining tractable sampling via an ordered reparameterization. Here we can fix a predefined causal ordering over the latent variables because the identifiability result determines the perturbation-responsive factors only up to permutation. We therefore choose a canonical ordering of the latent dimensions and enforce the corresponding triangular structure during learning, without loss of generality. This follows the standard practice in identifiable latent causal models (e.g., Liu et al. (2026c)). Algorithm 1 summarizes the training procedure, closely mirroring our implementation.

*Table 6.* Simulation hyperparameters.

| Hyperparameter | Value | Hyperparameter | Value |
|----------------|-------|----------------|-------|
| Batch size | 64 | $\mathbf{z}_\nu$ dim | 4 |
| Epochs | 100 | $\mathbf{z}_\iota$ dim | 7 |
| Learning rate | $1 \times 10^{-3}$ | $\beta_\nu$ | $1.5 \times 10^{-5}$ |
| $\beta_\iota$ | $5 \times 10^{-4}$ | $\alpha_{\text{contrast}}$ | 0.1 |

## H. Experimental Details

### H.1. Synthetic Data Experiments

**Data Generation.** We sample data following the data-generating process described in Sec. 4. The concrete simulation parameters are summarized in Table 5.

*Table 5.* Simulation data generation parameters.

| Quantity | Symbol | Value |
|----------|--------|-------|
| Observation dimension | $\mathbf{x}$ | 500 |
| Latent dimension (variant) | $\mathbf{z}_\nu$ | 4 |
| Latent dimension (invariant) | $\mathbf{z}_\iota$ | 7 |
| Intervention dimension | $\mathbf{u}$ | 12 |
| Training size | – | 3000 |
| Test size | – | 1000 |

**Training Setup and Hyperparameters.** We use the Adam optimizer with the hyperparameters listed in Table 6.

**Evaluation Metrics.** Identifiability of the variant block $\mathbf{z}_\nu$ is quantified using the mean correlation coefficient (MCC), which measures the one-to-one correspondence between each learned latent and its ground-truth. To compute MCC, we follow these steps:

1. **Compute correlation coefficients.** We first compute the pairwise correlation coefficients between the ground-truth latent components $\mathbf{z}_{\nu,i}$ and the learned latent components $\hat{\mathbf{z}}_{\nu,j}$ across the test samples. Specifically, for each pair $(i, j)$ we compute the Pearson correlation coefficient

$$\rho_{i,j} = \frac{\text{Cov}(\mathbf{z}_{\nu,i}, \hat{\mathbf{z}}_{\nu,j})}{\sigma_{\mathbf{z}_{\nu,i}} \, \sigma_{\hat{\mathbf{z}}_{\nu,j}}}, \tag{63}$$

   where the covariance and standard deviations are computed over the dataset. We take absolute values $|\rho_{i,j}|$ to account for the inherent sign ambiguity of latent variables.

2. **Solve the linear sum assignment problem.** Since the learned components may be permuted relative to the ground-truth latents, we solve a linear sum assignment problem to find the optimal one-to-one matching $\pi$ between ground-truth and learned components that maximizes the total absolute correlation:

$$\pi = \arg\max_{\pi \in \mathcal{S}_d} \sum_{i=1}^{d} |\rho_{i,\pi(i)}|, \tag{64}$$

   where $\mathcal{S}_d$ denotes the set of all permutations of $\{1, \ldots, d\}$ and $d$ is the latent dimension.

3. **Compute the mean correlation coefficient (MCC).** Given the optimal assignment $\pi$, we define the MCC as the average of the matched absolute correlations:

$$\text{MCC} = \frac{1}{d}\sum_{i=1}^{d}|\rho_{i,\pi(i)}|. \tag{65}$$

To assess block identifiability result over $\mathbf{z}_\iota$, we regress the ground-truth latent variables ($\mathbf{z}_\iota$) on their learned estimates ($\hat{\mathbf{z}}_\iota$), and report the coefficient of determination ($R^2$). High $R^2$ values close to one indicate successful block identifiability. To compute the coefficient of determination ($R^2$) for block-wise identifiability, we proceed as follows:

1. **Extract learned and ground-truth latents.** We collect the learned latent representations $\hat{\mathbf{z}}_\iota$ and the corresponding ground-truth latent variables $\mathbf{z}_\iota$ on the test set.

2. **Regression from learned to ground-truth latents.** Since identifiability is only defined up to an unknown transformation at the block level, we fit a regression model from the learned latents to the ground-truth latents. Concretely, we fit a function $f$ of the form

$$\mathbf{z}_\iota = f(\hat{\mathbf{z}}_\iota) + \epsilon. \tag{66}$$

3. **Compute the coefficient of determination.** After fitting the regressor, we compute the coefficient of determination

$$R^2 = 1 - \frac{\sum_{i=1}^{n}\|\mathbf{z}_{\iota,i} - \hat{\mathbf{z}}_{\iota,i}^{\text{pred}}\|^2}{\sum_{i=1}^{n}\|\mathbf{z}_{\iota,i} - \bar{\mathbf{z}}_\iota\|^2}, \tag{67}$$

where $\hat{\mathbf{z}}_{\iota,i}^{\text{pred}} = f(\hat{\mathbf{z}}_{\iota,i})$ is the predicted ground-truth latent for sample $i$, $\bar{\mathbf{z}}_\iota$ is the empirical mean of the ground-truth latents, and $n$ is the number of test samples.

4. **Interpretation.** A value of $R^2$ close to 1 indicates that the learned latent block can be (nonlinearly) transformed to accurately recover the true latent block, consistent with block-wise identifiability. Lower $R^2$ indicates loss of information or entanglement across blocks.

### H.2. Real-Data Experiments Details

**Dataset from (Norman et al., 2019)** For real-world perturbation data, we consider the large-scale Perturb-seq dataset from (Norman et al., 2019). It consists of 105,528 cells from an erythroleukemia cell line (K562) subjected to CRISPR activation (Gilbert et al., 2014) targeting 112 genes, resulting in 105 single-gene and 131 double-gene perturbation conditions. Each perturbation condition contains between 50 and 2,000 cells. Across all conditions, each cell is represented as a 5,000-dimensional vector $\mathbf{x}$, corresponding to the gene expression levels.

**Experimental Protocol for Comparing with Existing Latent Causal Models** In this setting, we follow the experimental protocol of Zhang et al. (2023). Specifically, we partition the dataset into training and test splits as follows. The training set consists of all unperturbed cells together with the 105 single-gene perturbation datasets $\mathcal{X}_1, \ldots, \mathcal{X}_{105}$. For each single-gene dataset containing more than 800 cells, we randomly hold out 96 cells to form a *single-gene test set*, while the remaining cells are used for training. The *double-gene test set* consists of the 112 double-gene perturbation datasets $\mathcal{X}_{106}, \ldots, \mathcal{X}_{217}$, which are entirely held out from training and used only for evaluation. This setup ensures that models are trained on unperturbed and single-gene perturbation data, but evaluated on both held-out single-gene cells and, more importantly, on combinatorial perturbations.

**Evaluation Metrics: RMSE and Population-Level $R^2$** In real single-cell perturbation datasets, the ground-truth latent variables are unobservable. We therefore report the root mean squared error (RMSE) between the predicted and observed mean expression vectors, which directly measures the absolute magnitude of prediction errors in gene expression space. In addition, $R^2$ cannot be used to assess latent recovery or identifiability, and is instead used as a measure of predictive utility at the level of gene expression. Concretely, for each perturbation condition, the model first generates a population of "virtual" cells conditioned on the perturbation label. We compute the mean gene expression vector of these generated cells and compare it to the mean expression vector of the experimentally observed cells under the same perturbation. A

linear regression is then fitted between the predicted and observed mean expression vectors, and the resulting coefficient of determination $R^2$ quantifies how well the model explains the population-level transcriptional response to the perturbation. Under this protocol, $R^2$ measures the accuracy of predicted perturbation effects in terms of explained variance rather than the recovery of latent causal variables, and therefore serves as a proxy for the practical usefulness of the learned representations in predicting gene expression changes under perturbations.

**Hyperparameter Settings for Real Data Experiments.** We use the Adam optimizer with hyperparameters detailed in Table 7.

*Table 7.* Real Data Hyperparameters.

| Hyperparameter | Value | Hyperparameter | Value |
|---|---|---|---|
| Batch size | 64 | Hidden dimension | 256 |
| Epochs | 100 | $\mathbf{z}$ dimension | 10, 35, 75, 105 |
| Learning rate | $1 \times 10^{-4}$ | $\alpha_{\text{contrast}}$ | 0.05 |
| $\beta_\nu, \beta_\iota$ | $1 \times 10^{-2}$ | | |

# I. Unperturbed Latent Subspace Analysis

In this section, we provide additional visual evidence that the invariant latent block $\mathbf{z}_\iota$ indeed captures perturbation-invariant background transcriptional programs, and does not encode perturbation-specific information.

We examine whether this invariance property holds globally across the full test set, including both single-gene and double-gene perturbations. Figure 6 shows t-SNE embeddings of $\mathbf{z}_\iota$ for all test samples under single-gene and double-gene perturbation conditions. In both cases, cells from different perturbation regimes remain well mixed and do not form separated clusters, suggesting that $\mathbf{z}_\iota$ generalizes its invariance property beyond the training distribution.

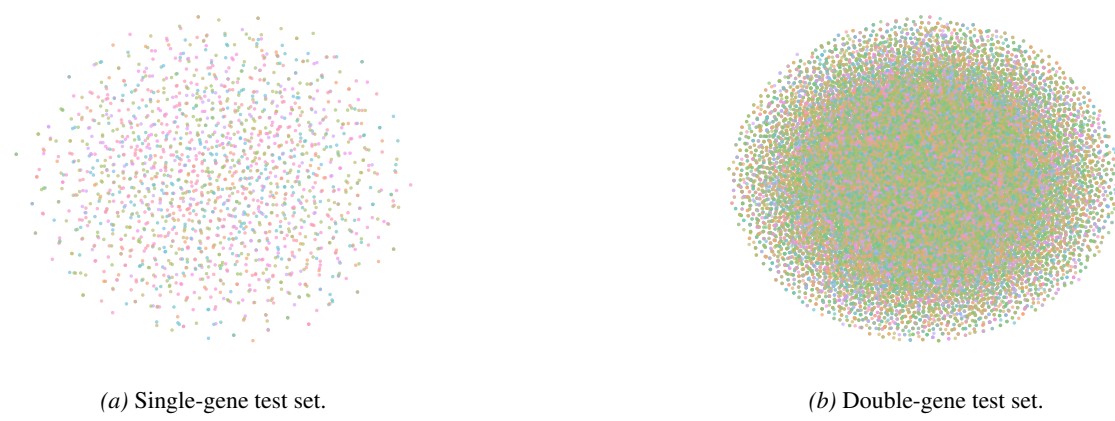

*(a)* Single-gene test set.      *(b)* Double-gene test set.

*Figure 6.* t-SNE visualization of the invariant block $\mathbf{z}_\iota$ for single-gene (a) and double-gene (b) perturbations in the test set.

Together, these results provide additional evidence that $\mathbf{z}_\iota$ captures perturbation-invariant variation in gene expression, validating the disentanglement between the invariant background block $\mathbf{z}_\iota$ and the perturbation-responsive block $\mathbf{z}_\nu$.

## J. Perturbed Latent Subspace Analysis

### J.1. The Learned Latent Causal Representations

In this section, we examine whether the latent causal representation learned by PerturbedVAE is (i) identifiable and consistently aligned with external perturbations, (ii) structured rather than dense or entangled, and (iii) biologically meaningful. We provide a sequence of visualizations and analyses to validate these properties.

**Alignment and Identifiability.** Following Zhang et al. (2023), we first present in Figure 7 the hit map between perturbed genes and the identifiable latent causal components $\mathbf{z}_\nu(i)$ learned by our model. Columns correspond to perturbed genes, while rows denote individual causal components. Each entry highlights the component most strongly associated with a given perturbation. This visualization assesses whether perturbations are consistently mapped to specific latent components, thereby validating both identifiability and alignment of the learned representation.

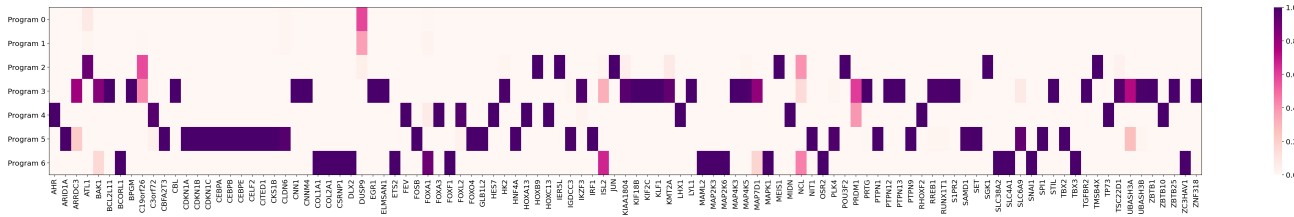

*Figure 7.* Perturbed gene hits on identifiable causal components.

**Learned Causal Structure.** To further examine the structure of the learned latent representation, we visualize the directed dependencies among the identifiable components $\mathbf{z}_\nu$. Figure 8 (left) shows the full adjacency matrix estimated by the model prior to thresholding, where color intensity reflects the signed strength of each estimated causal effect. For interpretability, we apply a threshold ($\tau = 0.25$) to prune weak connections, yielding a sparse graph that highlights the dominant causal relations (Figure 8, right). This comparison demonstrates that the learned structure is neither dense nor arbitrary, but exhibits a sparse and interpretable causal backbone.

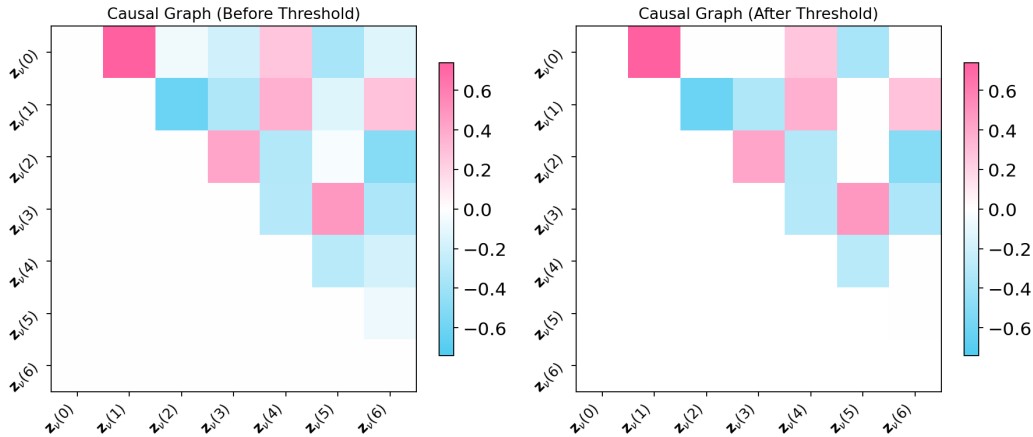

*Figure 8.* Visualization of the learned causal graph among identifiable components $\mathbf{z}_\nu$. **Left:** full adjacency matrix before thresholding. **Right:** sparse graph after thresholding ($\tau = 0.25$).

**Semantic Grounding of Latent Programs.** We next assess whether the latent components correspond to coherent biological programs. Figure 9 visualizes the inferred causal graph among latent components, where each node represents a latent program and each directed edge denotes an inferred causal dependency. To provide gene-level interpretability, we map each latent program back to its associated genes. The complete mapping is reported in Table 9, demonstrating that each latent component corresponds to a meaningful and coherent gene module.

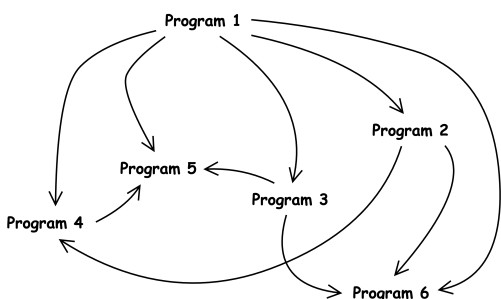

*Figure 9.* Inferred causal structure among latent programs.

**Biological Plausibility of Inferred Edges.**    Finally, we examine whether representative inferred causal edges are consistent with known biological mechanisms. Table 8 summarizes several directed edges together with their mechanistic interpretations and supporting references, illustrating that the learned structure aligns with established regulatory pathways rather than reflecting spurious statistical associations.

*Table 8.* Program-level representative edges: mechanistic rationale and supporting references.

| Edge | Mechanistic rationale (summary) | Refs. |
|---|---|---|
| DUSP9 → TGFBR2 | TGFBR2 activates ERK through a non-Smad branch; DUSP9 dephosphorylates ERK/JNK, attenuating this output. | (Emanuelli et al., 2008; Zhang, 2009) |
| DUSP9 → TP73 | c-Jun enhances TP73 stability; DUSP9 lowers JNK/ERK→AP-1 signaling, indirectly downregulating TP73. | (Koeppel et al., 2011; Emanuelli et al., 2008) |
| DUSP9 → CDKN1A | ERK→ELK1/EGR1 induces p21 transcription; DUSP9 suppresses ERK phosphorylation, blunting this induction. | (Lim et al., 1998; Ragione et al., 2003) |
| DUSP9 → SNAI1 | EMT induction requires SMAD3–AP-1 cooperation; DUSP9 attenuates AP-1, weakening SNAI1 transcription. | (Sundqvist et al., 2013; Fan et al., 2025) |
| JUN → TP73 | c-Jun stabilizes and potentiates TP73, enhancing apoptosis-related transcription. | (Koeppel et al., 2011) |
| JUN → SNAI1 | AP-1 cooperates with SMAD factors to elevate SNAI1 expression in TGF-$\beta$-driven EMT. | (Sundqvist et al., 2013; Fan et al., 2025) |
| TGFBR2 → CDKN1A | Canonical SMAD2/3/4 downstream of TGFBR2 transactivates p21, enforcing cytostasis. | (Ikushima & Miyazono, 2010) |

## J.2. Perturbation Information Preservation

In this section, we evaluate whether perturbation-specific information is preserved in the perturbed latent subspace rather than being suppressed by the invariant background representation. Following the main text, we use the top 20 differentially expressed (DE) genes as a perturbation-enriched readout to probe the preservation of perturbation-induced variation.

**Metric Definitions and Empirical Observations.**    To quantify information preservation, we compute performance metrics on two complementary feature sets for each perturbation condition:

- **All genes**: measurements computed using the entire 5,000-dimensional gene expression, reflecting the global state.

*Table 9.* Complete list of genes assigned to each latent program inferred from structure learning.

| Program | Genes |
|---|---|
| 1 | DUSP9 |
| 2 | ATL1, C19orf26, HOXB9, IER5L, JUN, MEIS1, POU3F2, SGK1, TMSB4X |
| 3 | ARRDC3, BAK1, BCL2L11, BPGM, CBL, CNN1, CNNM4, EGR1, ELMSAN1, HK2, IKZF3, KIAA1804, KIF18B, KIF2C, KLF1, KMT2A, LYL1, MAP4K3, MAP4K5, MAP7D1, PRDM1, PRTG, PTPN12, PTPN13, RREB1, RUNX1T1, S1PR2, STIL, TGFBR2, TSC22D1, UBASH3A, UBASH3B, ZBTB1, ZBTB25, ZNF318 |
| 4 | AHR, C3orf72, FEV, FOXA3, FOXL2, HES7, HOXA13, HOXC13, LHX1, MIDN, RHOXF2, TP73, ZBTB10 |
| 5 | ARID1A, CBFA2T3, CDKN1A, CDKN1B, CDKN1C, CEBPA, CEBPB, CEBPE, CELF2, CITED1, CKS1B, CLDN6, FOSB, FOXO4, GLB1L2, HNF4A, IGDCC3, IRF1, NIT1, PLK4, PTPN1, PTPN9, SAMD1, SET, SLC6A9, SPI1, TBX2 |
| 6 | BCORL1, COL1A1, COL2A1, CSRNP1, DLX2, ETS2, FOXA1, FOXF1, ISL2, MAML2, MAP2K3, MAP2K6, MAPK1, NCL, OSR2, SLC38A2, SLC4A1, SNAI1, TBX3, ZC3HAV1 |

- **DE genes**: measurements computed using the 20-dimensional sub-vectors corresponding to the top 20 most differentially expressed genes, which form a perturbation-enriched readout of the perturbed subspace.

We make the following empirical observations:

- **In-distribution (single-gene).** The model achieves high accuracy on both feature sets. The $R^2$ scores on the DE genes are nearly identical to those on all genes, while the RMSE on the DE subset is notably lower (Figure 11), indicating that perturbation-induced signals are well preserved under single-gene interventions.

- **Out-of-distribution (double-gene).** While the global $R^2$ on all genes remains high (around $\sim 0.98$), the $R^2$ on the DE genes exhibits a mild degradation for a subset of double-gene perturbations, with values in the $0.5$–$0.9$ range (Figure 10). This reflects the increased difficulty of zero-shot combinatorial extrapolation, where novel, potentially non-additive interactions must be inferred from single-gene training data.

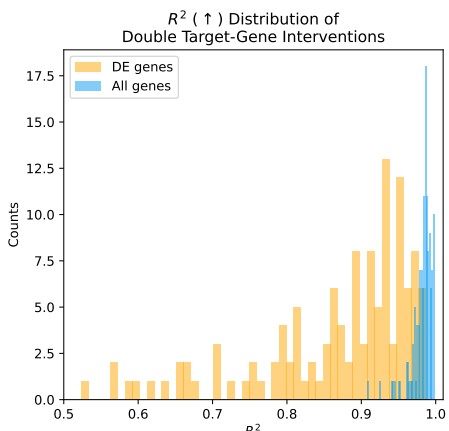
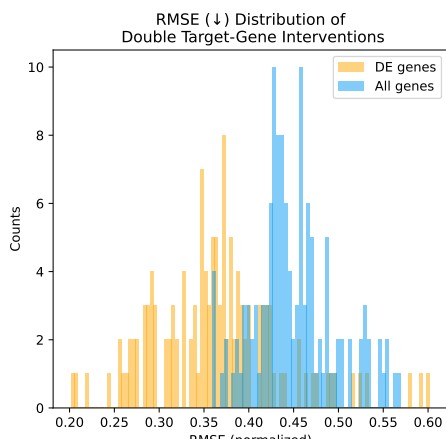

*Figure 10.* Performance on DE genes for double-gene perturbations.

**Interpretation: Information Preservation under Invariance.** These patterns are consistent with the intended design of the model: perturbation-invariant background variation is stabilized in the invariant block $\mathbf{z}_\iota$, while perturbation-specific information is retained in the perturbed block $\mathbf{z}_\nu$ rather than being suppressed.

**Preservation of Perturbation Signals.** The strong performance on DE genes in the single-gene setting indicates that the model successfully preserves perturbation-induced variation in $\mathbf{z}_\nu$ and propagates it to gene-level predictions.

**Limits under Combinatorial Generalization.** The moderate degradation of $R^2$ on DE genes for some double-gene perturbations reflects the intrinsic difficulty of zero-shot combinatorial causal prediction. Importantly, DE-gene RMSE typically remains low even when $R^2_{\mathrm{DE}}$ decreases, suggesting that the model often predicts the magnitude of key expression changes reasonably well, even when fine-grained variance patterns are harder to match.

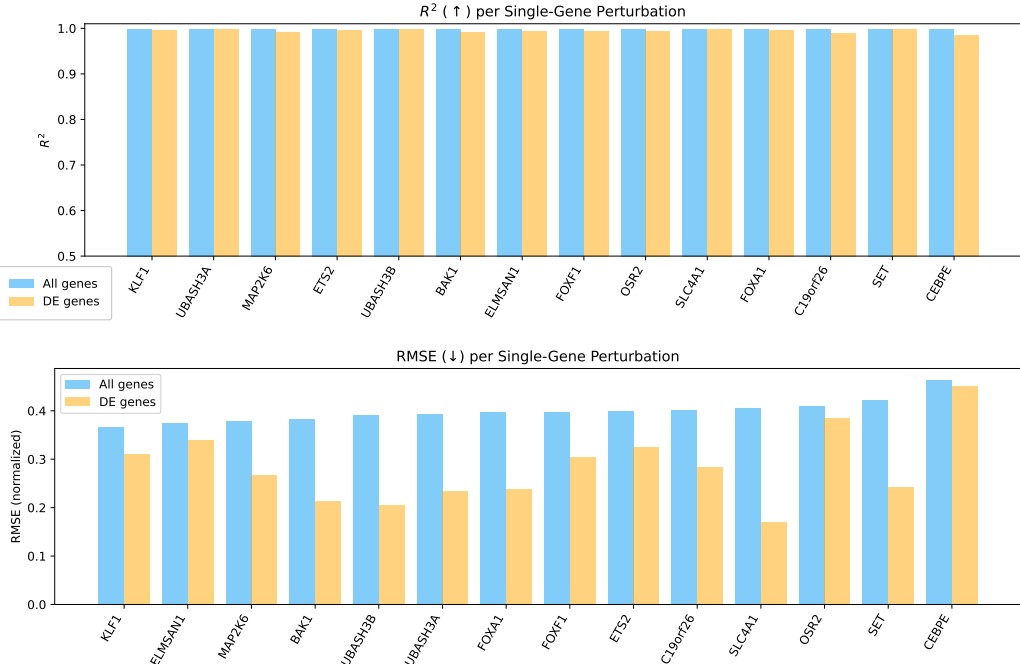

*Figure 11.* Performance on DE genes for single-gene perturbations.

## K. Cross-Dataset Single-Gene I.I.D. Check on `Replogle2022`

**Scope.** We include Replogle et al. (2022) as an additional cross-dataset single-gene i.i.d. check for the latent causal representation model comparison. This experiment evaluates whether the proposed perturbation-aware representation learning mechanism remains effective on an independent perturbation dataset. It is not intended to test double-gene OOD extrapolation, which is evaluated in the main Norman et al. (2019) combinatorial setting.

**Protocol.** We follow the same latent causal model comparison protocol as in the Norman et al. (2019) single-gene i.i.d. evaluation. For each method, we evaluate prediction quality at both the condition level and the cell level. At the condition level, we report L2 distance and $\Delta$Pearson between predicted and observed perturbation responses. At the cell level, we report RMSE. We additionally use $R^2$ as a sanity check to assess whether the predicted single-cell profiles preserve non-degenerate cell-level variation.

*Table 10.* `Replogle2022` single-gene i.i.d. evaluation for latent causal representation models. Condition-level metrics evaluate perturbation-level response recovery, while cell-level RMSE evaluates single-cell prediction fidelity.

| Method | L2 $\downarrow$ | $\Delta$Pearson $\uparrow$ | RMSE $\downarrow$ |
|---|---|---|---|
| SENA | $10.916 \pm 0.064$ | $0.057 \pm 0.007$ | $0.6761 \pm 0.0038$ |
| Discrepancy-VAE | $10.862 \pm 0.029$ | $0.056 \pm 0.006$ | $0.6731 \pm 0.0012$ |
| sVAE+ | $10.608 \pm 0.045$ | $0.136 \pm 0.002$ | $0.4905 \pm 0.0001$ |
| SAMS-VAE | $8.489 \pm 0.061$ | $0.157 \pm 0.014$ | $0.4818 \pm 0.0005$ |
| **PerturbedVAE** | $\mathbf{8.296 \pm 0.034}$ | $\mathbf{0.192 \pm 0.001}$ | $\mathbf{0.4815 \pm 0.0004}$ |

**Results.** As shown in Table 10, PerturbedVAE achieves the best condition-level performance among the compared latent causal representation models, with the lowest L2 error and the highest $\Delta$Pearson. It also obtains the lowest cell-level RMSE, indicating improved single-cell prediction fidelity on an independent perturbation dataset.

$R^2$ **Sanity Check.** Although not included in the table, we also examine population-level $R^2$ as a sanity check for preserving meaningful single-cell variation. All compared baselines yield negative cell-level $R^2$ values, indicating that their predicted cell-level profiles are worse than the empirical-mean predictor and suffer from severe distortion at the single-cell level. In contrast, PerturbedVAE is the only method that achieves a positive cell-level $R^2$, suggesting that the learned perturbation-aware representation better preserves cell-level structure while maintaining condition-level perturbation accuracy.

**Interpretation.** This cross-dataset result supports the representation-learning aspect of PerturbedVAE: explicitly separating perturbation-responsive variation from dominant invariant structure improves prediction not only on Norman et al. (2019), but also on an independent single-gene perturbation dataset. We emphasize that this experiment is complementary to the main Norman et al. (2019) double-gene OOD evaluation. The combinatorial generalization claim is supported by the Norman et al. (2019) double-gene benchmark, while Replogle et al. (2022) provides an additional i.i.d. check of representation learning robustness.

## L. Ablation Studies and Design Validation

This subsection reports a series of control and ablation experiments designed to isolate the contributions of key components of PerturbedVAE, including the choice of discrepancy metric, latent capacity allocation, and contrastive alignment.

**Control: MMD-Based Discrepancy Variant.** To rule out the possibility that the observed performance gains are simply due to using a different discrepancy loss, we evaluate a variant of our model in which the contrastive alignment term is replaced by a maximum mean discrepancy (MMD) regularizer, denoted as PerturbedVAE(MMD). This mirrors the MMD-based discrepancy formulation used in Zhang et al. (2023), enabling a direct, like-for-like comparison with Discrepancy-VAE under the same alignment criterion. This control therefore isolates whether the improvements arise from the proposed causal structure and disentanglement mechanism, rather than from the specific form of the discrepancy loss. Performance on single-gene perturbation prediction is reported in Table 11.

*Table 11.* Evaluation of the PerturbedVAE with MMD variant on single-gene perturbation prediction.

| Method | Metrics | | |
| --- | --- | --- | --- |
| | **RMSE** | $\mathbf{R}^2$ | **MMD** |
| **Discrepancy-VAE** (Zhang et al., 2023) | $0.5558_{\pm 0.0022}$ | $0.9916_{\pm 0.0014}$ | $0.3243_{\pm 0.0050}$ |
| **PerturbedVAE** (MMD) | $\mathbf{0.5485_{\pm 0.0013}}$ | $\mathbf{0.9958_{\pm 0.0003}}$ | $\mathbf{0.3077_{\pm 0.0036}}$ |

**Ablation on Latent Capacity Allocation.** We next ablate the allocation of latent capacity between the invariant block $\mathbf{z}_\iota$ and the perturbation-responsive block $\mathbf{z}_\nu$. As reported in Table 12, together with Figure 4, asymmetric allocations in which $\mathbf{z}_\iota$ is assigned substantially more capacity than $\mathbf{z}_\nu$ consistently outperform both equal-split and variant-heavy configurations on both in-distribution (single-gene) and out-of-distribution (double-gene) prediction. The invariant-heavy configuration $(z_\nu, z_\iota) = (20, 85)$ achieves the lowest RMSE and highest $R^2$ across settings, indicating that sufficient capacity for modeling background transcriptional programs is critical for accurate generalization.

In contrast, when $\mathbf{z}_\iota$ is under-resourced (e.g., $(85, 20)$ or $(50, 55)$), performance degrades noticeably and becomes largely indistinguishable across such settings. This suggests that (i) the perturbation-responsive subspace $\mathbf{z}_\nu$ is already adequate at relatively small dimensionalities, and increasing its capacity yields diminishing returns; whereas (ii) the invariant block $\mathbf{z}_\iota$ constitutes the primary performance bottleneck.

*Table 12.* Results on single- and double-gene perturbations under different capacity allocations of $\mathbf{z}_\nu$ and $\mathbf{z}_\iota$.

| Dimension | Single-Gene Perturbation | | Double-Gene Perturbation | |
| --- | --- | --- | --- | --- |
| | **RMSE** | $\mathbf{R}^2$ | **RMSE** | $\mathbf{R}^2$ |
| $\mathbf{z}_\nu = \mathbf{z}_\iota$ | $0.4084_{\pm 0.0011}$ | $0.9875_{\pm 0.0007}$ | $0.4627_{\pm 0.0003}$ | $0.9649_{\pm 0.0003}$ |
| $\mathbf{z}_\nu > \mathbf{z}_\iota$ | $0.4084_{\pm 0.0010}$ | $0.9875_{\pm 0.0007}$ | $0.4627_{\pm 0.0002}$ | $0.9649_{\pm 0.0002}$ |
| $\mathbf{z}_\nu < \mathbf{z}_\iota$ | $\mathbf{0.3995_{\pm 0.0013}}$ | $\mathbf{0.9977_{\pm 0.0002}}$ | $\mathbf{0.4474_{\pm 0.0007}}$ | $\mathbf{0.9865_{\pm 0.0009}}$ |

**Ablation on Contrastive Alignment.** Finally, we ablate the contrastive alignment term by comparing PerturbedVAE with and without the alignment loss ($\alpha = 0.05$ vs. $\alpha = 0$) under a fixed total latent dimensionality ($z = 105$). As shown in Table 13, removing the alignment loss leads to consistent performance degradation on both single- and double-gene prediction.

Empirically, when $\alpha = 0$, the invariant block $\mathbf{z}_\iota$ collapses and carries little information (with $\mathrm{KL}_\iota \to 0$), causing the effective representation to be dominated by the perturbation-responsive block $\mathbf{z}_\nu$. As a result, performance resembles that of capacity splits with $\mathbf{z}_\nu \geq \mathbf{z}_\iota$, in which the model effectively ignores the invariant subspace. In contrast, with alignment enabled, $\mathbf{z}_\iota$ remains informative and stable, preventing leakage of perturbation-specific effects into the invariant block and yielding substantially better generalization, particularly on out-of-distribution double-gene perturbations.

These results indicate that contrastive alignment is a key mechanism for sustaining the informativeness of the invariant block and maintaining the intended disentanglement, and is therefore critical for the robust performance of PerturbedVAE.

*Table 13.* Single- and double-gene performance under contrastive alignment ablation.

| Contrastive Alignment | Single-Gene Perturbation | | Double-Gene Perturbation | |
|:---:|:---:|:---:|:---:|:---:|
| | RMSE | $R^2$ | RMSE | $R^2$ |
| ✗ | $0.4083_{\pm 0.0011}$ | $0.9875_{\pm 0.0007}$ | $0.4626_{\pm 0.0002}$ | $0.9650_{\pm 0.0002}$ |
| ✓ | $\mathbf{0.3995_{\pm 0.0013}}$ | $\mathbf{0.9977_{\pm 0.0002}}$ | $\mathbf{0.4474_{\pm 0.0007}}$ | $\mathbf{0.9865_{\pm 0.0009}}$ |

## M. Perspective on Additive Linear Baselines for Double-Gene Perturbation

This section details the main-paper discussion on the classical additive linear baseline for combinatorial perturbation prediction. We aim to clarify (i) why additive models can be highly competitive on standard pseudobulk benchmarks on Norman2019, as also observed by Ahlmann-Eltze et al. (2025), and (ii) what aspects of the prediction problem are not well captured by purely additive or regression-centric objectives. We then report supplementary results that compare Perturbed-VAE with the additive baseline and GEARS under a strict single-gene $\rightarrow$ double-gene OOD protocol, using evaluation criteria that separately probe average responses and single-cell heterogeneity.

**Why Additive Baselines Can Be Strong on `Norman2019`.** Recent benchmarking results (Ahlmann-Eltze et al., 2025) highlight an important empirical characteristic of Norman2019: when evaluated on condition-level pseudobulk responses (i.e., averages over cells), even sophisticated models (including GEARS (Roohani et al., 2024) and several foundation-model variants) may not consistently outperform a simple additive predictor under squared-error metrics. A plausible explanation is that, for many gene pairs and for many high-expression targets, the dominant component of the double-perturbation response is well approximated by a near-linear superposition of single-gene effects. In such regimes, the additive baseline benefits from a strong inductive bias that is directly aligned with the benchmark objective.

**Perspective: Latent Causal Modeling Beyond Pseudobulk Regression.** Our work targets a different modeling goal from regression-centric predictors that directly map perturbation labels to pseudobulk profiles. We aim to learn a structured latent causal representation that supports mechanism-level disentanglement and generalization to combinatorial interventions without using any double-perturbation supervision during training. Concretely, we model single-cell observations as arising from low-dimensional latent causal variables $\mathbf{z}$ whose dynamics are modulated by interventions $\mathbf{u}$ and corrupted by biologically meaningful stochasticity $\mathbf{n}$. In this formulation, $\mathbf{z}$ is not gene expression itself; rather, it can be interpreted as latent cellular programs, pathway activities, or regulatory modules that mediate perturbation effects. The observed expression $\mathbf{x}$ is treated as a nonlinear projection of these latent factors through the VAE decoder.

The explicit noise term $\mathbf{n}$ reflects substantial cell-to-cell stochasticity in single-cell transcriptomics (e.g., transcriptional bursting and technical variability) that is typically suppressed by pseudobulk averaging. Modeling this stochasticity, rather than collapsing it, is intended to help separate perturbation-driven signals from unstructured variation and to make the learned latent mechanisms more suitable for downstream interpretation and OOD generalization. This perspective motivates evaluating models not only on average effects, but also on how well they capture perturbation-conditioned single-cell variability.

**Feature Space and Evaluation Granularity.** To balance standard comparability with causal/biological validity, we report results on two complementary gene sets and at two levels of evaluation granularity.

**Gene sets.** (i) **High-expression benchmark subset:** following Ahlmann-Eltze et al. (2025), we compute metrics on the 1,000 most highly expressed genes in control cells. This subset provides a high–signal-to-noise regime and reflects the commonly used pseudobulk benchmark setting. (ii) **Genome-wide profile:** we also evaluate on the full set of 5,000 genes. This is more aligned with the design goal of PerturbedVAE, since background cellular programs may manifest as subtle, distributed signals that are not restricted to high-expression genes and may be missed by top-expression filtering.

**Evaluation levels.** (i) **Condition-level pseudobulk:** we average single-cell profiles within each perturbation condition to form a pseudobulk vector and report standard metrics (Delta Pearson[2], $L_2$, RMSE, $R^2$). These metrics quantify how well a method recovers the average conditional response associated with each perturbation and provide a direct comparison

---

[2]We report *Pearson Delta* at the pseudobulk level, defined as the Pearson correlation (across genes) between predicted and observed perturbation-induced changes relative to control.

to additive baselines. (ii) **Perturbation-conditioned single-cell evaluation:** for each perturbation label $\mathbf{u}$, the model produces a predicted mean expression vector, interpreted as a deterministic summary of $p_\theta(\mathbf{x} \mid \mathbf{u})$. We then compare this predicted mean against the ensemble of observed single-cell profiles under the same $\mathbf{u}$, computing RMSE and $R^2$ at the single-cell level and finally averaging across held-out double-perturbation conditions. Unlike pseudobulk metrics, this evaluation probes how well the model explains perturbation-conditioned variability in single-cell states. In PerturbedVAE, strong performance here is consistent with the intended disentanglement: $\mathbf{z}_\iota$ captures perturbation-invariant background programs, while $\mathbf{z}_\nu$ captures perturbation-responsive mechanisms that reshape the single-cell expression landscape (see also App. J.2 for DE-gene–enriched probes).

*Table 14.* Supplementary evaluation on genome-wide expression profiles (single-gene $\rightarrow$ double-gene OOD). ***Note.*** A dash (–) indicates that the corresponding cell-level $R^2$ is negative (worse than a trivial condition-mean predictor) and is therefore not informative for comparing methods in this setting.

| Method | Condition-level | | | | Cell-level | |
|---|---|---|---|---|---|---|
| | Prediction error ($L2$) | Delta Pearson | RMSE | $R^2$ | RMSE | $R^2$ |
| **Additive** | $2.5407_{\pm 0.0000}$ | $0.9076_{\pm 0.0000}$ | $0.0887_{\pm 0.0000}$ | $0.6431_{\pm 0.0000}$ | $0.4424_{\pm 0.0000}$ | $-$ |
| **GEARS** | $4.6797_{\pm 0.2620}$ | $0.4631_{\pm 0.0644}$ | $0.1514_{\pm 0.0086}$ | $0.9730_{\pm 0.0032}$ | $0.5861_{\pm 0.0031}$ | $-$ |
| **PerturbedVAE** | $3.7238_{\pm 0.0012}$ | $0.6869_{\pm 0.0005}$ | $0.1285_{\pm 0.0015}$ | $0.9965_{\pm 0.0005}$ | $0.4494_{\pm 0.0008}$ | $0.9840_{\pm 0.0011}$ |

*Table 15.* Supplementary evaluation on the high-expression gene subset (single-gene $\rightarrow$ double-gene OOD).

| Method | Condition-level | | | | Cell-level | |
|---|---|---|---|---|---|---|
| | Prediction error ($L2$) | Delta Pearson | RMSE | $R^2$ | RMSE | $R^2$ |
| **Additive** | $2.4906_{\pm 0.0000}$ | $0.9101_{\pm 0.0000}$ | $0.0870_{\pm 0.0000}$ | $0.6470_{\pm 0.0000}$ | $0.4332_{\pm 0.0000}$ | $-$ |
| **GEARS** | $4.2649_{\pm 0.2044}$ | $0.5068_{\pm 0.0710}$ | $0.1381_{\pm 0.0065}$ | $0.9682_{\pm 0.0065}$ | $0.5746_{\pm 0.0018}$ | $-$ |
| **PerturbedVAE** | $3.6491_{\pm 0.0010}$ | $0.6936_{\pm 0.0004}$ | $0.1259_{\pm 0.0013}$ | $0.9951_{\pm 0.0005}$ | $0.4411_{\pm 0.0007}$ | $0.9758_{\pm 0.0011}$ |

**Discussion.** Tables 14-15 summarize results for PerturbedVAE, the additive baseline, and GEARS under the strict single-gene $\rightarrow$ double-gene OOD protocol. Among the deep learning models, PerturbedVAE attains higher $R^2$ and lower RMSE than GEARS on both gene sets, suggesting that conditioning prediction on a learned causal latent representation can support stronger OOD generalization than directly learning a perturbation-to-expression mapping with the graph neural network baseline.

Consistent with Ahlmann-Eltze et al. (2025), the additive baseline remains highly competitive on condition-level pseudobulk metrics, achieving the lowest $L_2$ error and highest Delta Pearson, especially on the high-expression subset on which the benchmark is commonly defined. This behavior is expected when the dominant component of the average response is approximately linear and weakly interacting, matching the inductive bias of the additive model. In such settings, more flexible models must recover this near-linearity from data while also accommodating residual non-additive effects, which can lead to a small gap on average-effect metrics even when the model is useful for other aspects of the problem.

However, condition-level metrics alone can obscure differences in how methods relate to single-cell variability. In our evaluation, both the additive baseline and GEARS yield negative cell-level $R^2$ on held-out double perturbations, indicating that their predicted condition means do not improve over a trivial condition-mean predictor when assessed against the distribution of single-cell profiles. In contrast, PerturbedVAE achieves substantially higher cell-level $R^2$ while remaining competitive on pseudobulk metrics. This pattern is consistent with the modeling objective of PerturbedVAE: learning disentangled latent factors that capture both shared background programs and perturbation-responsive mechanisms, thereby providing a coherent generative account of how perturbations reshape single-cell state distributions. From a CRL perspective, such single-cell–level fidelity is particularly relevant for downstream tasks such as mechanism interpretation, causal structure discovery, and robust OOD generalization, including settings where linear compositionality assumptions may not hold.

## N. Supplementary PCA/scVI-Based Additive Controls

Our main comparison with simple baselines follows the additive linear model emphasized by recent large-scale benchmarking work (Ahlmann-Eltze et al., 2025). In addition, related benchmarking studies have reported that simple representation-extractor pipelines, such as PCA-based representations combined with linear predictors, can be competitive in single-cell perturbation prediction (Bendidi et al., 2024). We therefore include PCA/scVI-based additive controls as supplementary analyses. These experiments test whether generic low-dimensional representation extraction followed by additive extrapolation is sufficient for the double-gene OOD setting considered in this work.

We evaluate two representation-extractor controls: PCA+Additive and scVI+Additive. PCA provides a linear low-dimensional representation of the expression matrix, while scVI provides a generic nonlinear variational representation. In both cases, the extracted representation is used with the same additive prediction pipeline. These controls differ from PerturbedVAE in that they do not explicitly separate perturbation-invariant background variation from perturbation-responsive effects, nor do they learn a perturbation-conditioned latent mechanism for unseen combinatorial interventions.

*Table 16.* Supplementary PCA/scVI-based additive controls for double-gene OOD prediction. PCA and scVI are used as generic representation extractors followed by an additive prediction pipeline.

| Method | RMSE $\downarrow$ | $R^2 \uparrow$ |
|---|---|---|
| PCA + Additive | $0.4907 \pm 0.0002$ | – |
| scVI + Additive | $0.4735 \pm 0.0002$ | – |
| **PerturbedVAE** | $\mathbf{0.4474 \pm 0.0007}$ | $\mathbf{0.9865 \pm 0.0009}$ |

As shown in Table 16, PCA+Additive and scVI+Additive provide competitive simple controls, confirming that generic low-dimensional representations can capture useful structure for perturbation prediction. However, both remain below PerturbedVAE in double-gene OOD prediction. These supplementary results do not diminish the value of simple PCA-based pipelines; rather, they clarify their role as controls for generic representation compression. In our double-gene OOD setting, generic compression followed by additive extrapolation remains insufficient to match PerturbedVAE, supporting the need for perturbation-aware modeling beyond standard low-dimensional representation extraction.

