# OpenReview forum: "What Makes a Representation Good for Single-Cell Perturbation Prediction?"
_ICML.cc/2026/Conference — ICML 2026 regular_

### Official Review · Reviewer_QrL3 · 2026-03-03

**Soundness:** 2
**Presentation:** 3
**Significance:** 3
**Originality:** 4
**Overall Recommendation:** 4
**Confidence:** 4

**Summary:**

This paper investigates representation learning for single-cell perturbation prediction and proposes the Perturbation Suppression Hypothesis, which argues that dominant perturbation-invariant signals can obscure sparse perturbation-specific effects in gene expression data, limiting generalization. To address this, the authors introduce PerturbedVAE, a perturbation-aware variational autoencoder that explicitly separates perturbation-invariant and perturbation-responsive latent variables via structured variational inference and a contrastive alignment objective. The perturbation-responsive block is modeled with a latent causal structure, and the authors provide an identifiability analysis under partial intervention assumptions. Experiments on the Norman Perturb-seq dataset show improved performance over foundation models, simple baselines, and prior causal VAE methods, particularly for combinatorial perturbation prediction, along with interpretable latent structures.

**Compliance With Llm Reviewing Policy:**

Affirmed.

**Key Questions For Authors:**

Q1. Quantitative support for the Perturbation Suppression Hypothesis

In Section 2.1, the paper argues that perturbation-specific signals occupy only a small fraction of gene expression variation. Would it be possible to provide a quantitative analysis to support this claim (e.g., proportion of differentially expressed genes per perturbation, variance explained by perturbation labels, or effect size distributions)? If such analysis confirms strong sparsity, it would substantially strengthen the empirical grounding of the hypothesis and increase my confidence in the motivation.

Q2. Realism of the assumptions in Theorem 1

The assumptions underlying Theorem 1 (e.g., invertibility, optimal alignment, sufficient environmental diversity) appear quite strong, especially in the context of real single-cell transcriptomic data. In scenarios where perturbations induce large-scale state transitions (e.g., differentiation or cell-type changes), achieving a perfectly invariant block may be unrealistic. Could the authors provide empirical evidence that the Norman dataset (or other datasets) approximately satisfies these conditions? Alternatively, could they clarify how sensitive the method is when these assumptions are only partially met?

Q3. Evaluation on additional datasets

The current experimental validation focuses mainly on the Norman dataset, which involves a relatively homogeneous cell type. Have the authors evaluated PerturbedVAE on additional perturbation datasets (e.g., involving different cell types or perturbation modalities)? Results on more diverse datasets would help assess the robustness and general applicability of the proposed framework. Positive evidence here could substantially strengthen the empirical contribution.

Q4. Scope of generalization evaluation

The paper emphasizes improved generalization, but it seems that they only evaluate primarily the “seen gene but unseen pair” scenario . Have the authors tested other generalization regimes, such as unseen genes, unseen pairs, or cross-cell-type transfer (as done in works like GEARS)?

**Limitations:**

The authors discuss some modeling assumptions in the theoretical section, but there is no dedicated limitations discussion. It would strengthen the paper to explicitly acknowledge the strength of the identifiability assumptions, the limited dataset diversity in experiments, and the potential risks of over-interpreting learned latent causal structures in biological applications. A short dedicated limitations section would improve transparency.

**Strengths And Weaknesses:**

Strengths

1.Presentation

Overall, the paper is clearly written and easy to follow. The narrative (from introducing the Perturbation Suppression Hypothesis to proposing PerturbedVAE and providing theoretical and empirical support) is logically structured. Although I primarily work in bioinformatics and am not deeply specialized in causal representation learning or variational inference, I found the theoretical section reasonably accessible. The authors do a good job explaining the modeling assumptions and connecting them to the architecture design.

2.Originality

The idea of explicitly separating perturbation-invariant and perturbation-responsive latent variables is interesting and conceptually appealing. While the individual components (VAE, structured SEM, alignment loss) are not entirely new, combining them in a perturbation-aware framework is a thoughtful and creative design choice. Framing the problem through the Perturbation Suppression Hypothesis provides a useful perspective on why standard representations might struggle with perturbation prediction.

3. Interpretability

The interpretability analysis, especially the program-level causal graph visualization (Figure 5), is particularly interesting from a biological standpoint. As someone working in computational biology, I appreciate seeing models that attempt to recover structured and biologically meaningful patterns rather than focusing solely on predictive accuracy.


Weaknesses

1. Limited empirical scope

The experimental evaluation is conducted primarily on a single realworld dataset (Norman et al.), which involves a relatively homogeneous cell type. While the results are promising, it is difficult to assess how broadly the proposed framework generalizes across datasets with different cell types, perturbation magnitudes, or noise characteristics. The limited empirical diversity constrains the strength of the paper’s generalization claims.

2. Interpretation of foundation model comparison

In Table 2, simple classical models (e.g., ElasticNet, RandomForest) perform competitively and even outperform foundation models. This raises the possibility that the dataset may not strongly benefit from high-capacity pretrained representations. As a result, it is not entirely clear whether the observed improvements stem from addressing a fundamental suppression issue, or whether the task complexity is already manageable with simpler models.

3. Some conceptual ambiguity in the prediction pipeline

While reading the theoretical and modeling sections, I was somewhat confused about the prediction mechanism. The task is to predict perturbed expression given unperturbed expression and a perturbation condition. Conceptually, this does not necessarily require encoding the post-perturbation expression during inference. In Section 3.2, the inference model is defined, but the test-time counterfactual prediction pipeline is not described very clearly. A more explicit explanation of how latent variables are constructed and used at prediction time would improve clarity and reproducibility.

---

> ### Author Rebuttal · Authors · 2026-03-30
>
> We thank the reviewer for these helpful suggestions.
>
> `Q1: Quantitative support.` Thanks for this important suggestion! We conduct a quantitative analysis on the Norman et al. (2019) dataset:
>
> | Statistic | Value |
> |---|---:|
> | DE genes (%), all non-control perturbations| 1.07±0.98 |
> | DE genes (%), single-gene perturbations only| 0.61±0.77 |
> | Gene-wise ANOVA $\eta^2/R^2$ | 1.64±3.89 |
> | abs($log_2 FC$) P90 / P99 | 0.077 / 0.533 |
> | Mean raw count, background vs responsive genes | 50.10 vs 5.18 |
>
> These results show that perturbation-specific signals are highly sparse: only a very small fraction of genes are differentially expressed, the variance explained by perturbations is low, and most effect sizes are concentrated near zero. Moreover, perturbation-responsive genes tend to have substantially lower expression levels compared to background genes, indicating that perturbation effects are weak relative to dominant background variation.
>
> Together, these observations provide quantitative support for the hypothesis.
>
> `Q2:  The assumptions in Theorem 1.`  We agree that Theorem 1 is an idealized identifiability result. As in most identifiability analyses in causal representation learning, we introduce a set of assumptions, since the problem is in general unidentifiable without certain assumptions. Most of these assumptions are adopted from prior work, as discussed in Appendix Sec. E.
>
> In practice, indeed, these assumptions may be only approximately satisfied. Real single-cell datasets (e.g., Norman et al., 2019) may exhibit partial violations, especially under stronger perturbations or state transitions. However, our method does not rely on exact satisfaction of these conditions. Instead, the latent decomposition and alignment objective act as a soft structural bias, encouraging separation between dominant background variation and perturbation-induced effects even under imperfect conditions.
>
> Empirically, this robustness is further supported by our ablations: removing the alignment objective (Table 11) or altering latent capacity (Tables 3 and 10) degrades performance, indicating that the proposed structure remains beneficial beyond idealized settings. We agree that further empirical verification of these assumptions would strengthen the practical grounding, and will clarify this point in the revision.
>
> We will make this point more clear, thanks again!
>
>
> `Q3: Evaluation on additional datasets.`
> Please refer to `Q1 to Reviewer TxTn`.
>
> `Q4: Scope of generalization evaluation.` There are different generalization regimes in this problem. Combinatorial generalization (e.g., double-gene OOD) might be more tractable, as the model observes single-gene perturbations during training and might generalize by composing these effects. Intuitively, if the effects of single-gene perturbations span a sufficiently rich space, their combinations might be approximated by composing these learned effects. In contrast, unseen-gene generalization is more challenging, as it requires extrapolating beyond the observed perturbations. In practice, this typically requires additional structure or side information (e.g., gene embeddings or biological priors) to provide a basis for why such generalization is possible.
>
> Given this above, we focus on combinatorial generalization under partial interventions. We agree that extending to unseen-gene or cross-cell-type generalization is an important direction for future work.
>
> `Additional clarifications.`
> * **Foundation-model comparison:** FMs do not necessarily outperform simpler baselines at the **individual-cell level**, where sparse perturbation-specific signals are mixed with substantial cell-to-cell variability and technical noise. In this setting, pretrained representations often capture broad population-level structure better, while simpler baselines can be more robust under low signal-to-noise conditions.  Although we briefly discussed this in **Compared With Simple Baselines**, we agree that it deserves a deeper treatment and will expand this discussion in the revision.
>
> * **Clarity of the prediction pipeline:** At inference time, the model does not require post-perturbation expression as input. Instead, it takes an **unperturbed/control cell** together with a perturbation-condition vector $\mathbf{u}$, and predicts the perturbed expression through the learned perturbation-conditioned latent mechanism. In our setting, training uses only single-gene perturbations, so $\mathbf{u}$ has a single active entry during training; at test time, the same learned mapping is directly evaluated on an unseen two-hot perturbation vector for a held-out double-gene combination. We will release an easy-to-follow codebase to support reproducibility and follow-up research.
>
>
> ----
>
> Thanks again for the insightful suggestions, which will help us further improve the paper in the revision.

---

> > ### Author Rebuttal · Reviewer_QrL3 · 2026-04-02
> >
> > The supplemented experiments are enough to justify the effectiveness of the method. Since most of my concerns have been addressed, I would like to raise the score by 1.

---

> > > ### Author Response · Authors · 2026-04-02
> > >
> > > We sincerely thank the reviewer for the time and thoughtful review, the reviewer's positive assessment of our work is greatly encouraging to us.

---

### Official Review · Reviewer_t34o · 2026-03-06

**Soundness:** 3
**Presentation:** 3
**Significance:** 3
**Originality:** 2
**Overall Recommendation:** 4
**Confidence:** 3

**Summary:**

The author develops VAEs that disentangle perturbation effect from intrinsic variation. A contrastive learning architecture is used to minimize the distance between perturbed and unperturbed cells in the latent space representing intrinsic biological variations. The author also provides additional identifiability results. This is a novel topic that could have high impact on the tasks of perturbation prediction in single-cell genomics.

**Compliance With Llm Reviewing Policy:**

Affirmed.

**Final Justification:**

The rebuttal has addressed my main concerns by following up with more ablation studies of including results in comparison to random shuffling. Also, the authors compared to other standard VAE approaches which I previously missed.

**Key Questions For Authors:**

1. In real data, if you reshuffle the control label randomly with other perturbed labels, does the model performance decrease?
2. In figure 5, could you get the same results using a standard VAE such as scVI?
3. How did you pick the run to show for figure 5? (It is understandable that a run with better fit is chosen.)
4. Could you be more clear in the main body in terms of the number of genes in the expression matrix of the real data used for training and evaluation?
5. In Table 2, could you also compare the random? Say treat the control group cell distribution as your naive prediction.


I am happy to increase the score of the questions are properly addressed.

**Limitations:**

yes

**Strengths And Weaknesses:**

**Strength:**

Soundness:

1. Somewhat thorough evaluation and benchmarking.
2. Used both real data and simulated data.
3. Provided an identifiability proof.

The paper is well presented.

Significance: disentangling real perturbation effects from intrinsic variation could have profound impact in the field of single-cell genomics.

The paper is original in its methodology and application.

**Weakness:**
1. Would be nice if there were more ablation studies. (See questions.)

---

> ### Author Rebuttal · Authors · 2026-03-30
>
> Thank you for these helpful suggestions.
>
> `Q1: Effect of random reshuffling.` Thanks for this insightful question. To directly address it, we conducted a reshuffle experiment, where the control samples are randomly replaced by samples from other perturbed conditions. The results are:
>
> | Method | RMSE | $R^2$ |
> |---|---:|---:|
> | PerturbedVAE (reshuffle) | _0.4507±0.0005_ | _0.9820±0.0007_ |
> | PerturbedVAE (w/o Align) | 0.4626±0.0002 | 0.9650±0.0002 |
> | PerturbedVAE | **0.4474±0.0007** | **0.9865±0.0009** |
>
> Reshuffling leads to a mild degradation compared to the original control-based alignment, but still clearly outperforms w/o Align.
>
> This is consistent with the role of the alignment term. In Eq. (5), the ELBO part is unaffected by the anchor choice; the difference arises from the alignment term (Eq. (4)), which encourages the invariant block to absorb shared/background variation, thereby freeing the perturbation-responsive block to capture perturbation-induced effects.
>
> Under reshuffling, the anchor is no longer a true control, but it still provides a relative regularization signal, partially enforcing structure across samples. This explains why it remains beneficial compared to removing alignment. However, perturbed anchors may contain perturbation-specific effects, making them a noisier and less reliable reference for isolating invariant structure. Consequently, reshuffling helps, but is weaker than control-based alignment.
>
> Overall, this indicates that the gain from alignment is not tied to a specific control label, but to the inductive bias it introduces for separating dominant shared variation from perturbation-responsive variation. At the same time, true controls provide the cleanest biological reference, and thus achieve the best performance.
>
> `Q2: scVI.` A standard VAE such as **scVI** dose not parameterizes or constrains a directed dependency structure among latent variables, and is therefore not designed to recover a sparse latent **DAG** over programs as in our framework. More importantly, such models do not provide identifiability guarantees: even if another method happened to output a similar estimated graph, that agreement alone would not justify a causal interpretation, since it could still be one observationally compatible solution among many. This is not merely a theoretical concern: without identifiability, the learned representations may entangle distinct underlying factors, leading to poor interpretability.
>
> To make this more direct, we additionally included experiments using **scVI** and **PCA** as representation extractors followed by an **additive model** (over 5 random seeds). This provides a clearer empirical contrast and further supports the point that our method learns a more task-aligned and interpretable latent structure, rather than merely another generic low-dimensional embedding.
>
> | Method | $RMSE$ | $R^2$ |
> | -------- | -------- | -------- |
> | PCA + Additive     | $0.4907_{\pm 0.0002}$     |$-$ |
> | scVI + Additive | $0.4735_{\pm 0.0002}$ | $-$ |
> | PerturbedVAE     | $\mathbf{0.4474_{\pm 0.0007}}$     | $\mathbf{0.9865_{\pm 0.0009}}$|
>
>
> `Q3: Run selection for Figure 5.` Figure 5 was not selected as the single best-looking run. We trained the model with five random seeds, and only genes that were **stably assigned to the same learned programs across seeds** were shown in Figure 5. Some genes do vary in their program assignment across seeds (Table 8 shows the assigment under 1 seed), which is expected because our method learns structure at the latent program level, while gene-level assignment is a downstream projection based on the learned soft weights (converted to hard assignment by maximum weight). Thus, Figure 5 is intended to visualize the cross-seed stable and interpretable part of the learned structure.
>
> `Q4: Feature dimensionality of the real data.` The raw dataset contains approximately 19,264 genes, while the real-data experiments use a preprocessed **5,000-HVG expression matrix** (currently described in Appendix H.2). This feature space is not restricted to the 112 perturbed genes, but covers a much broader transcriptomic feature space. We will revise this accordingly. Thanks again!
>
> `Q5: Naive control baseline.` We implement a naive baseline that uses the control-group distribution as the prediction in both the single gene IID and double gene OOD settings (over five random seeds). This provides a sanity-check reference and confirms that our model improves over simply reverting to the control state.
>
> | Method | Single Gene I.I.D |  | Double Gene O.O.D |  |
> |---|---:|---:|---:|---:|
> |  | RMSE | $R^2$ | RMSE | $R^2$ |
> | Naive Baseline | 0.5569±0.0013 | 0.9862±0.0006 | 0.5941±0.0002 | 0.9643±0.0001 |
> | PerturbedVAE | **0.3995±0.0013** | **0.9977±0.0002** | **0.4474±0.0007** | **0.9865±0.0009** |
>
>
> Thanks again for the constructive suggestions, which have substantially enriched our manuscript. We will incorporate the relevant additional experiments accordingly.

---

> > ### Author Rebuttal · Reviewer_t34o · 2026-04-01
> >
> > I thank the reviewer for the replies. Most of my concerns have been addressed. Therefore, I am moving the score up by 1.

---

> > > ### Author Response · Authors · 2026-04-02
> > >
> > > We sincerely thank the reviewer for the time and thoughtful feedback. The comments have helped us improve the quality and clarity of the manuscript.

---

### Official Review · Reviewer_TxTn · 2026-03-09

**Soundness:** 1
**Presentation:** 2
**Significance:** 2
**Originality:** 3
**Overall Recommendation:** 4
**Confidence:** 5

**Summary:**

The authors propose PerturbedVAE, a latent causal generative model designed to predict single-cell gene expression responses to genetic perturbations. The paper introduces the "Perturbation Suppression Hypothesis," arguing that sparse perturbation-specific signals are often overwhelmed by dominant, perturbation-invariant background cellular programs. To address this, PerturbedVAE uses a contrastive alignment loss against unperturbed control cells to explicitly force the invariant latent block to capture background programs, freeing a variant block to learn causal perturbation effects. The model is evaluated on the Norman et al. (2019) dataset , where the authors claim it outperforms large foundation models (FMs) and older causal baselines on zero-shot combinatorial predictions.

**Compliance With Llm Reviewing Policy:**

Affirmed.

**Final Justification:**

I am improving my review score by 2 points as the author have completely resolved two of of my concerns (comparison against STATE and TxPert and comparison against scVI and PCA). My other concern on limited benchmarking datasets is only partially resolved, as while the authors have added Replogle 2022 in the rebuttal which strengthens the submission, I encourage them to include even more benchmark datasets to make the claims of the paper more robust and generalizable.

**Key Questions For Authors:**

- Dataset Scale: Norman (2019) is a relatively small dataset (~112 targeted genes). Can you provide evaluation results on genome-scale screens, such as Replogle (2022) or Nadig (2024), to prove your model's contrastive alignment scales effectively?

- Benchmarking Literature: How do you address the findings in recent benchmarking papers (e.g., Bendidi 2024 [1]) that show simple linear methods and PCA often outperform complex deep learning models on these tasks? Why was this literature not cited or used to establish trivial baselines?

[1] Benchmarking Transcriptomics Foundation Models for Perturbation Analysis: one PCA still rules them all, Bendidi I et al, 2024, https://arxiv.org/pdf/2410.13956

**Limitations:**

No. While the authors discussed theoretical limitations regarding their identifiability assumptions in Appendix B, they did not adequately discuss the massive empirical limitations of validating their framework on a single dataset without genome-scale verification, nor did they acknowledge the existence of much stronger concurrent baselines.

**Strengths And Weaknesses:**

#### Soundness

- Strengths: The theoretical framing is elegant. By explicitly modeling "partial interventions" (where only a few genes are perturbed while the rest remain invariant), the authors bridge a critical gap between theoretical causal representation learning and biological reality.

- Weaknesses : The empirical evaluation is fundamentally inadequate for an ICML 2026 submission.

    - Single, Outdated Dataset: The model is evaluated on exactly one real-world dataset (Norman 2019). Norman is small by modern standards (targeting only ~112 genes). Validating zero-shot combinatorial prediction requires genome-scale screens like Replogle (2022) or Nadig (2024). Testing on Norman alone makes it impossible to assess if the contrastive alignment holds up in the noisy, high-dropout regime of genome-scale data.
    - Hidden Linear Baselines: Recent literature has shown that simple additive linear baselines perform exceptionally well on the Norman benchmark. The authors acknowledge this but relegate the additive linear baseline to Appendix L, rather than putting it in the main text (Table 2).
    - Missing Simple Baselines: The paper lacks direct predictive comparisons against trivial baselines, such as "mean control gene expression" or PCA-based regression. PCA is only used as a diagnostic linear probe, not as a predictive baseline.

#### Presentation

- Strengths: The paper is well-written, and the "Perturbation Suppression Hypothesis" is an intuitive, well-articulated concept that clearly motivates the architecture.

- Weaknesses: The presentation is structurally deceptive. The presentation is structurally deceptive and fails to properly contextualize the work within the literature. First, by framing generic Foundation Models as the primary competitors in the main text while burying the strongest linear baseline in the appendix, the paper artificially inflates the perceived performance gap. Second, the authors completely fail to cite and discuss crucial contemporary competitors. State-of-the-art models specifically engineered for transcriptomic perturbation prediction, such as TxPert and STATE, are entirely missing from the references.


#### Significance

- Strengths: The core problem : predicting combinatorial out-of-distribution biological responses, is highly relevant to drug discovery and systems biology.

- Weaknesses: The paper fails to benchmark against the actual state-of-the-art. By completely ignoring powerful recent models explicitly designed for transcriptomic perturbation prediction (such as TxPert and STATE), the paper's significance is severely diminished. Beating older causal models and generic FMs on a small 2019 dataset does not advance the 2026 state-of-the-art.

#### Originality

- Strengths: Combining contrastive learning, VAEs, and causal discovery to specifically counter perturbation suppression is a creative and biologically motivated application of existing deep learning components.

---

> ### Author Rebuttal · Authors · 2026-03-30
>
> We thank the reviewer for the comments.
>
> `Q1: Norman (2019) is a relatively small.` Thank you for the comment.
>
> Our work focuses on leveraging the generalization ability of causal modeling to improve perturbation prediction, and therefore we primarily target OOD evaluation settings. In particular, double-gene OOD prediction is a representative and challenging task that directly aligns with our goal.
>
> Norman (2019) is one of the few datasets that contains systematic double-gene perturbations, and has been widely adopted in prior work (e.g., Discrepancy-VAE, GEARS).
>
> The above motivates our initial focus on the double-gene OOD setting.
>
> To further address the concern on dataset scale and robustness, we additionally conduct experiments on the larger Replogle (2022) dataset to validate the generality of our method. The results are shown below:
>
> | Method | Condition level |  | Cell level |  |
> |---|---:|---:|---:|---:|
> |  | L2 | ΔPearson | RMSE | $R^2$ |
> | SENA | 10.916±0.064 | 0.057±0.007 | 0.6761±0.0038 | - |
> | Discrepancy-VAE | 10.862±0.029 | 0.056±0.006 | 0.6731±0.0012 | - |
> | sVAE+ | 10.608±0.045 | 0.136±0.002 | 0.4905±0.0001 | - |
> | SAMS-VAE | 8.489±0.061 | 0.157±0.014 | 0.4818±0.0005 | - |
> | PerturbedVAE | **8.296±0.034** | **0.192±0.001** | **0.4815±0.0004** | **+** |
>
>
> `Q2: Benchmarking Literature Bendidi 2024.`  Thank you for raising this important point. In our initial submission, we focused on the most recent benchmarking study by Ahlmann-Eltze et al. (2025) [1], which provides a careful analysis of the limitations of many deep learning approaches relative to simple linear/additive models. This work informed both the positioning of our method and our consideration of simple baselines. As a result, we did not explicitly cite Bendidi et al. (2024), which we agree is also a valuable and relevant contribution. We will include it in the revised version and appropriately discuss its findings.
>
> Moreover, to address this concern, we explicitly include simple baselines based on PCA and scVI combined with additive prediction, as shown below:
>
> | Method | RMSE | $R^2$ |
> |---|---:|---:|
> | PCA + Additive | 0.4907±0.0002 | - |
> | scVI + Additive | 0.4735±0.0002 | - |
> | PerturbedVAE | **0.4474±0.0007** | **0.9865±0.0009** |
>
> Our results suggest that, when the task requires disentangling invariant background structure from perturbation-responsive effects, structured latent causal models such as ours may provide advantages in such settings.
>
>
> `Q3: Hidden Linear Baselines, the presentation is structurally deceptive.` **We take the additive baseline very seriously.** Following recent benchmarking work (e.g., Ahlmann-Eltze et al., 2025), we explicitly considered additive linear baselines in our evaluation.
>
> Rather than mixing it directly with other baselines in Table 2, we chose to highlight it separately in the main text under “Perspective on the Additive Linear Baseline” (Lines 376–384, left column; Lines 341–350, right column). Our intention was to provide a more focused discussion of its behavior and implications, rather than treating it as just another entry in the table.
>
> Due to its importance, we also provide a detailed analysis, with nearly two full pages of additional discussion in Appendix L. Given the page limit of the submission, these extended results and analyses were moved to the appendix.
>
> That said, we agree with your point that including the additive baseline directly in Table 2 may be also a reasonable presentation choice. In the revised version, we will include its results in Table 2 for comparison.
>
> `Q4: Comparison with Txpert and State.` Our CRL-based method primarily targets the individual-cell level, which is generally more challenging than population-level prediction. While population-level effects are aggregated from individual cells, strong performance at the population level may not necessarily translate to accurate modeling at the single-cell level.
>
> In this regime, foundation models do not necessarily have an advantage, as individual-cell prediction is more sensitive to noise and cell-to-cell variability. This makes it crucial to explicitly model structured and perturbation-specific variation, rather than relying on global representations optimized for aggregated signals.
>
> We provide a comparison below:
>
> | Method | RMSE | $R^2$ |
> |---|---:|---:|
> | Txpert | 0.4892±0.0442 | 0.9306±0.0590 |
> | State | 0.4981±0.0046 | 0.9475±0.0021 |
> | PerturbedVAE | **0.4474±0.0007** | **0.9865±0.0009** |
>
>
> [1] Ahlmann-Eltze C, Huber W, Anders S. Deep-learning-based gene perturbation effect prediction does not yet outperform simple linear baselines[J]. Nature Methods, 2025, 22(8): 1657-1661.
>
> ----
>
> The main concerns raised (dataset scale, missing baselines, and comparison to recent models) are addressed through additional experiments and clarifications. We hope these results help clarify the validity and scope of our conclusions, and most importantnly, encourage a re-evaluation. Thanks again!

---

> > ### Author Rebuttal · Reviewer_TxTn · 2026-04-01
> >
> > I am improving my review score by 2 points as the author have completely resolved two of of my concerns (comparison against STATE and TxPert and comparison against scVI and PCA). My other concern on limited benchmarking datasets is only partially resolved, as while the authors have added Replogle 2022 in the rebuttal which strengthens the submission, I encourage them to include even more benchmark datasets to make the claims of the paper more robust and generalizable.

---

> > > ### Author Response · Authors · 2026-04-02
> > >
> > > We thank the reviewer for the thoughtful comments and valuable suggestions, these have helped us substantially improve the clarity and quality of the manuscript.

---

### Official Review · Reviewer_UeKS · 2026-03-11

**Soundness:** 3
**Presentation:** 3
**Significance:** 3
**Originality:** 3
**Overall Recommendation:** 4
**Confidence:** 2

**Summary:**

The paper studies single-cell perturbation-effect prediction and argues that most variation in gene expression is perturbation-invariant, whereas perturbation-specific effects are relatively sparse, a phenomenon the authors refer to as the perturbation suppression hypothesis. Motivated by this, the paper proposes PerturbedVAE, a variational framework that separates perturbation-invariant and perturbation-responsive latent variables. The model is trained with an ELBO objective together with a contrastive regularizer that encourages the invariant latent variables to remain aligned across perturbed and control conditions. The paper further presents an identifiability analysis for a linear latent causal model and specifies assumptions under which the perturbation-responsive structure can be recovered from partial-interventional data. Empirically, the authors evaluate the method on synthetic and real-world datasets, showing competitive or improved performance relative to baselines and suggesting that the perturbation-responsive latent variables capture interpretable structure.

**Compliance With Llm Reviewing Policy:**

Affirmed.

**Final Justification:**

The rebuttal clarified my questions, but my concern regarding generalizability remains. While the authors claim in the paper that their approach enables principled generalization, they later state that “generalization should be understood within a specific regime.” However, this regime is not characterized rigorously (e.g., through the identifiability conditions), and the authors empirically evaluate only combinations of seen perturbations. Thus, it remains unclear to the reader under which regime the method is actually expected to generalize. That said, I think the overall idea is interesting and the empirical results are convincing. Therefore, I will keep my score at “weak accept”.

**Key Questions For Authors:**

1.	How is the perturbation variable $ \mathbf{u} $ instantiated in the synthetic and real-data experiments, and how are perturbation-dependent quantities parameterized as functions of $ \mathbf{u} $? In particular, the paper suggests that $ \mathbf{u} $ is represented as an identifier such as a one-hot encoding, but it is then unclear how the model can generalize to genuinely novel perturbations outside the observed set. Relatedly, could the authors clarify how the intervention-dependent quantities in the theoretical model, such as $ \lambda_{\nu\iota}(\mathbf{u}) $, $ \lambda_{\nu\nu}(\mathbf{u}) $, $ \mu_\nu(\mathbf{u}) $, and $ \beta_\nu(\mathbf{u}) $, are specified or parameterized in the synthetic and real-world experiments?
2.	How should practitioners choose the dimensions of the perturbation-invariant and perturbation-responsive latent spaces?
3.	What are concrete biological example of what should count as perturbation-invariant information with respect to the proposed framework?
4.	Is the perturbation-invariant / perturbation-responsive split intended to be intrinsic to the biological system, or only relative to (a class of) perturbations observed during training? What happens if an unseen perturbation affects parts of the perturbation-invariant latent variables (as the perturbation is novel/ potentially targets new things as unseen during training).

**Limitations:**

yes

**Strengths And Weaknesses:**

## Strengths
- PerturbedVAE addresses the important task of single-cell perturbation-effect prediction while aiming to retain interpretability, which is valuable for biological applications where understanding perturbation-driven structure matters in addition to predictive accuracy.
- The paper proposes perturbation suppression hypothesis as a potential cause for existing approaches (i.e., foundation models and CRL) to show suboptimal perturbation response predictions.
- The paper includes a rigorous identifiability analysis for when perturbation-responsive latent variables can be recovered under the assumed latent causal model, which adds a principled theoretical component beyond a purely empirical method.
- The empirical study is thorough, combining synthetic and real-world benchmarks, diverse baseline comparisons, ablations and sensitivity analyses, and evaluation in practically important out-of-distribution settings. On the reported metrics, PerturbedVAE is competitive with or outperforms the compared baselines across several settings.

## Weaknesses
- The paper treats the split into perturbation-invariant and perturbation-responsive factors as if it were a stable property of the data-generating process, but in practice that split may only be meaningful relative to the family of perturbations observed during training. A feature that looks invariant for one perturbation set may become responsive once a new, stronger, or qualitatively different perturbation is introduced. The notion of “invariant” may be perturbation-class-dependent, and the paper does not make that dependence fully explicit.
- Relatedly, the role of the perturbation variable $\mathbf{u}$ remains somewhat unclear. If $\mathbf{u}$  is represented only as a one-hot identifier of a perturbation, it is difficult to see how the method can generalize to genuinely novel perturbations beyond the observed set, rather than learning a perturbation-specific factorization tied to the training distribution.
- The current evaluation focuses on RMSE and  $R^2$, which primarily assess average prediction accuracy and may not reflect whether the model captures the full single-cell perturbational distribution. Additional distributional metrics such as Wasserstein distance or MMD could provide a more complete evaluation.

### Minor Comments
- It would be helpful to include additional architectural details of PerturbedVAE, either in the main text or appendix, to make the experimental implementation easier to follow.
- A running biological example illustrating what should count as perturbation-invariant versus perturbation-responsive signal would help motivate the latent-variable separation more concretely.

---

> ### Author Rebuttal · Authors · 2026-03-30
>
> Thanks for the reviewer's comments.
>
> `Q1: Perturbation representation u & OOD generalization.` In our experiments, u is represented as an observed perturbation-condition vector (one-hot), which is mapped through a neural network to parameterize intervention-dependent quantities  (e.g., $\lambda_{\nu\iota}(\mathbf{u})$, $\lambda_{\nu\nu}(\mathbf{u})$, $\mu_{\nu}(\mathbf{u})$, and $\beta_{\nu}(\mathbf{u})$). This allows $\mathbf{u}$ to modulate the perturbation-responsive latent subspace, including both the noise distributions and causal relationships.
>
> Following previous work, e.g. Zhang et al., NeurIPS 2023, we consider a changeling double-gene OOD setting: training uses single-gene perturbations, while testing involves unseen combinations (two-hot vectors) of genes observed during training. For example, the model is trained on [0,1,0] and [0,0,1], and evaluated on their unseen combination [0,1,1]. In this setting, generalization arises from applying a shared u-conditioned mechanism to new combinations, rather than from an gene-composition rule.
>
> `Q2: Latent dimension selection.`  We view the latent split as reflecting the structure of the problem, but a useful practical default is to allocate more dimensions to the perturbation-invariant block $z_{\iota}$ than to the perturbation-responsive block $z_{\nu}$. The intuition is that the latter $z_{\nu}$ captures perturbation-specific, identifiable effects, which are typically sparse, while the former $z_{\iota}$ models broader background variation and thus requires larger capacity. This is consistent with the imbalance highlighted by the perturbation suppression hypothesis.
>
> This design is also supported by our ablations (Appendix K / Table 10), where settings with $z_{\nu} < z_{\iota}$ perform better. In practice, we recommend starting with $z_{\nu} < z_{\iota}$ and selecting the final dimensions based on validation performance and stability.
>
> `Q3: Biological example of invariant information.` In our framework, perturbation-invariant information refers to the part of the representation that captures background variation and remains relatively stable across the perturbations considered in latent spcae. When mapped back to the observation space, this would typically correspond to broad **background cellular programs** shared across perturbed and control cells.
>
> Concrete biological examples include **housekeeping-like programs** (e.g., core translation, cytoskeletal maintenance, or basic metabolic activity) and, in many settings, relatively stable aspects of **cell identity/state** that are preserved across the perturbations under study. By contrast, perturbation-responsive information captures the residual expression changes specifically induced by the perturbation, such as shifts in downstream target programs or pathway-specific transcriptional responses.
>
>
> `Q4: Relativity and robustness of the invariant/response split.` Thanks for such insightful comments!
>
> We first clarify that the invariant / responsive decomposition in our framework is not intended to be an intrinsic, globally fixed property of the biological system, but is defined relative to the available interventional data. In other words, under different observed intervention sets, the resulting invariant and responsive subspaces may differ. This is consistent with intuition: as the diversity of observed interventions increases, more variation can be attributed to perturbation-specific effects.
>
> That said, we emphasize that the goal of our framework is to introduce such a decomposition, which is adapted to the scope of the available interventional data.
>
> Regarding unseen perturbations (e.g., new single-gene interventions not observed during training), this is an interesting and important setting. In our current framework, such perturbations fall outside the training intervention distribution, and handling them typically requires additional structure or side information (e.g., gene-level features or pathway information), which we do not assume in this work.
>
> Nevertheless, our formulation has the potential to extend to such settings. In particular, incorporating new perturbations can be viewed as expanding the intervention set and updating the learned decomposition accordingly, which is closely related to continual or online learning settings. Designing mechanisms to efficiently integrate new perturbations into the learned latent structure is an interesting direction for future work.
>
> `Additional clarifications.`
> * **Additional metrics.** We already discussed this point in Appendix K by including an MMD-based variant of PerturbedVAE. As shown in Table 9, PerturbedVAE (MMD) improves over the compared CRL baseline.
> * **Reproducibility.** We note that the Appendix. H already provides pseudocode and the main hyperparameter settings. In the revision, we will release an easy-to-follow codebase to support reproducibility and follow-up research.

---

> > ### Author Rebuttal · Reviewer_UeKS · 2026-04-02
> >
> > Thanks for answering my questions. I think an important limitation to make more explicit is that, as noted in the answer, the perturbation-invariant / perturbation-responsive split is defined relative to the observed intervention set, and the theoretical scope is likewise restricted to perturbations seen during training. Thus, it is unclear how the guarantees extend to unseen perturbations, even for combinations of observed single-gene perturbations. The empirical performance suggests that the approach remains effective in the specific unseen settings considered, namely combinations of seen perturbations, but a more direct treatment of this setting, as mentioned in the answer (e.g., via continual or online learning, or additional structural or side information), remains open. I think this limitation is especially worth stating clearly given that part of the motivation is that existing methods can _yield non-generalizable predictors_, while the paper positions the proposed approach as _enabling principled generalization to unseen perturbations_.

---

> > > ### Author Response · Authors · 2026-04-03
> > >
> > > Thanks for the reviewer's additional feedback.
> > >
> > > Generalization should be understood within a specific regime; without such a context, one may not expect a model to generalize to arbitrary settings (i.e., there is no free lunch). In this work, we focus on out-of-distribution combinatorial generalization, i.e., predicting unseen combinations of perturbations that are individually observed during training, as emphasized throughout the paper.
> > >
> > > The setting of unseen single-gene interventions you mentioned, corresponds to a different and more challenging generalization regime. From a causal modeling perspective, principled generalization typically relies on a given underlying causal graph, this is guaranteed. In contrast, unseen single-gene interventions here involve perturbations that are not covered by the intervention on the given causal structure, effectively requiring extrapolation beyond the learned causal structure. This would therefore require additional assumptions or mechanisms. For example, such generalization would generally require richer perturbation descriptors than a one-hot identity, such as GO-based embeddings, or pathway/ontology features, which are outside the scope of the current work.
> > >
> > > We will clarify this more explicitly, and add a discussion in the revision, thanks a lot!

---

### Decision · Program_Chairs · 2026-04-30

**Decision:**

Accept (regular)

**Comment:**

The paper identifies a genuine and underappreciated problem (perturbation signal suppression), proposes a principled and well-motivated solution, provides identifiability theory, and demonstrates strong empirical results that survived scrutiny during a substantive rebuttal. All four reviewers recommend acceptance (three raised their scores post-rebuttal), and the remaining concerns are scoping and presentation issues addressable in the camera-ready. Therefore, I recommend acceptance.

The authors should ensure the revision: (1) explicitly scopes the generalization claims to the combinatorial OOD regime and discusses limitations for unseen single-gene perturbations; (2) incorporates all rebuttal experiments into the main paper or appendix; (3) includes the additive linear baseline in Table 2; and (4) clarifies the test-time prediction pipeline more explicitly in the main text.